# A Truncated Newton Method for Optimal Transport

**Mete Kemertas**[1,4*]       **Amir-massoud Farahmand**[2,3,1]       **Allan D. Jepson**[1]

[1]University of Toronto, [2]Polytechnique Montréal, [3]Mila - Quebec AI Institute, [4]Vector Institute

## Abstract

Developing a contemporary optimal transport (OT) solver requires navigating trade-offs among several critical requirements: GPU parallelization, scalability to high-dimensional problems, theoretical convergence guarantees, empirical performance in terms of precision versus runtime, and numerical stability in practice. With these challenges in mind, we introduce a specialized truncated Newton algorithm for entropic-regularized OT. In addition to proving that locally quadratic convergence is possible without assuming a Lipschitz Hessian, we provide strategies to maximally exploit the high rate of local convergence in practice. Our GPU-parallel algorithm exhibits exceptionally favorable runtime performance, achieving high precision orders of magnitude faster than many existing alternatives. This is evidenced by wall-clock time experiments on 24 problem sets (12 datasets $\times$ 2 cost functions). The scalability of the algorithm is showcased on an extremely large OT problem with $n \approx 10^6$, solved approximately under weak entropic regularization.

## 1 Introduction

The optimal transportation problem has long been a cornerstone of various disciplines, ranging from physics (Bokanowski & Grébert, 1996; Léonard, 2012; Levy et al., 2021) to machine learning and computer vision (Ferns et al., 2004; Pitie et al., 2005; Gulrajani et al., 2017; Genevay et al., 2018). Traditional approaches (Pele & Werman, 2009; Lee & Sidford, 2014), while exact and theoretically robust, encounter significant computational hurdles in high-dimensional settings. The (re-)introduction of entropic regularized OT (EOT), as pioneered by Cuturi (2013), has mitigated challenges in scalability by regularizing the classical problem, thereby enabling solutions via the GPU-friendly Sinkhorn-Knopp matrix scaling algorithm. This advancement has yielded substantial speed improvements, making it several orders of magnitude faster in high dimensions than traditional solvers. However, EOT methods necessitate a delicate balance between regularization strength and convergence speed, a trade-off that can compromise the precision of the solution.

Despite significant recent progress towards improving this trade-off, many state-of-the-art solvers still struggle to outperform aggressively tuned Sinkhorn iterations in practice (Jambulapati et al., 2019; Lin et al., 2019). While they offer superior theoretical guarantees, their practical performance is often less compelling, particularly in terms of speed and scalability. Existing algorithms either suffer from high computational complexity or fail to leverage modern hardware capabilities, such as GPU parallelization, effectively. To bridge this gap, we develop a new algorithm that remains numerically stable and converges rapidly even at extremely weak regularization levels, thereby enhancing precision in practice. By simultaneously exploiting the inherent parallelism of GPUs and superlinear local convergence of truncated Newton algorithms, our method scales effortlessly to high-dimensional problems, offering a pragmatic yet theoretically sound solution to the OT problem.

Our contributions are as follows: **(i)** we develop a specialized (linear) conjugate gradient algorithm for obtaining an approximation of the Newton direction for the EOT dual problem and analyze its convergence properties, **(ii)** we use the approximate (truncated) Newton direction in conjunction with a helper routine to develop a solver for the EOT dual problem and prove its superlinear local convergence, as well as per iteration computational cost, **(iii)** we develop an adaptive temperature annealing approach, based on the MDOT framework of Kemertas et al. (2025), to maximally

---

*Correspondence to: kemertas@cs.toronto.edu. Code available at github.com/metekemertas/mdot_tnt.

exploit this fast local rate, and finally **(iv)** present compelling empirical results via wall-clock time benchmarking in a GPU setting against a large suite of alternative algorithms in $n = 4096$ dimensions.

## 2 BACKGROUND AND RELATED WORK

**Notation and Definitions.** In this work, we are concerned with discrete OT. $\Delta_n \subset \mathbb{R}^n_{\geq 0}$ denotes the $(n-1)$-simplex. The row sum of an $n \times n$ matrix $P$ is given by $\boldsymbol{r}(P) := P\mathbf{1}$ and the column sum by $\boldsymbol{c}(P) := P^\top \mathbf{1}$. Given target marginals $\boldsymbol{r}, \boldsymbol{c} \in \Delta_n$, the transportation polytope is written as $\mathcal{U}(\boldsymbol{r}, \boldsymbol{c}) = \{P \in \mathbb{R}^{n \times n}_{\geq 0} \mid \boldsymbol{r}(P) = \boldsymbol{r}, \boldsymbol{c}(P) = \boldsymbol{c}\}$. Division, $\exp$ and $\log$ over vectors or matrices indicate element-wise operations. Vectors in $\mathbb{R}^n$ are taken to be column vectors and concatenation of two column vectors $\boldsymbol{x}, \boldsymbol{y}$ is $(\boldsymbol{x}, \boldsymbol{y})$. Elementwise minimum and maximum of a vector $\boldsymbol{x}$ is written as $\boldsymbol{x}_{\min}$ and $\boldsymbol{x}_{\max}$. Matrix and vector inner products alike are given by $\langle \cdot, \cdot \rangle$. An $n \times n$ diagonal matrix with $\boldsymbol{x} \in \mathbb{R}^n$ along the diagonal is written as $\mathbf{D}(\boldsymbol{x})$. We write $\chi^2(\boldsymbol{y}|\boldsymbol{x})$ for the $\chi^2$-divergence given by $\langle \boldsymbol{x}, (\boldsymbol{y}/\boldsymbol{x})^2 \rangle - 1 \geq \|\boldsymbol{y} - \boldsymbol{x}\|_1^2$. For the square root of $\chi^2(\boldsymbol{y}|\boldsymbol{x})$, we write $\chi(\boldsymbol{y}|\boldsymbol{x})$ with a slight abuse of notation. The Shannon entropy of $\boldsymbol{r} \in \Delta_n$ is denoted $H(\boldsymbol{r}) = -\sum_i r_i \log r_i$. We write $D_{\mathrm{KL}}(\boldsymbol{x}|\boldsymbol{y}) = \sum_i x_i \log(x_i/y_i) + \sum_i y_i - \sum_i x_i$ for the KL divergence between $\boldsymbol{x}, \boldsymbol{y} \in \mathbb{R}^n_{>0}$. We denote LogSumExp reductions along the rows and columns of $X$ by $\mathrm{LSE}_r(X) := \log\big(\exp\{X\}\mathbf{1}\big)$ and $\mathrm{LSE}_c(X) := \log\big(\exp\{X^\top\}\mathbf{1}\big)$.

### 2.1 OPTIMAL TRANSPORT AND ENTROPIC REGULARIZATION

We study the discrete optimal transport problem, formulated as the following linear program:

$$\underset{P \in \mathcal{U}(\boldsymbol{r}, \boldsymbol{c})}{\text{minimize}} \quad \langle P, C \rangle, \tag{1}$$

where we assume the $n \times n$ cost matrix has entries $C_{ij} \in [0, 1]$. Cuturi (2013) re-popularized EOT, showing that entropic regularization can help quickly approximate the solution of (1) on GPUs:

$$\underset{P \in \mathcal{U}(\boldsymbol{r}, \boldsymbol{c})}{\text{minimize}} \quad \langle P, C \rangle - \frac{1}{\gamma} H(P), \tag{2}$$

where the regularization weight $\gamma^{-1} \in \mathbb{R}_{>0}$ is called the *temperature*. It can be shown with ease that since the objective in (2) is strictly convex in $P$, problem (2) has a unique solution of the form

$$P(\boldsymbol{u}, \boldsymbol{v}; \gamma) = \exp\{\boldsymbol{u}\mathbf{1}^\top + \mathbf{1}\boldsymbol{v}^\top - \gamma C\}. \tag{3}$$

Using the form of the solution of (2), the following unconstrained dual problem can be solved instead:

$$\underset{\boldsymbol{u}, \boldsymbol{v} \in \mathbb{R}^n}{\text{minimize}} \quad g(\boldsymbol{u}, \boldsymbol{v}; \gamma) = \sum_{ij} P(\boldsymbol{u}, \boldsymbol{v}; \gamma)_{ij} - 1 - \langle \boldsymbol{u}, \boldsymbol{r} \rangle - \langle \boldsymbol{v}, \boldsymbol{c} \rangle, \tag{4}$$

where we keep the constant $-1$ as a convention. Solving (4) given initial $\boldsymbol{u}, \boldsymbol{v}$ amounts to a *Bregman projection* onto $\mathcal{U}(\boldsymbol{r}, \boldsymbol{c})$ in the sense that $P(\boldsymbol{u}^*, \boldsymbol{v}^*) = \arg\min_{P \in \mathcal{U}(\boldsymbol{r}, \boldsymbol{c})} D_{\mathrm{KL}}(P|P(\boldsymbol{u}, \boldsymbol{v}))$ (Kemertas et al., 2025). Noting that $\nabla_{\boldsymbol{u}} g = \boldsymbol{r}(P) - \boldsymbol{r}$ and $\nabla_{\boldsymbol{v}} g = \boldsymbol{c}(P) - \boldsymbol{c}$, we write:

$$\nabla^2 g = \begin{pmatrix} \mathbf{D}(\boldsymbol{r}(P)) & P \\ P^\top & \mathbf{D}(\boldsymbol{c}(P)) \end{pmatrix}_{2n \times 2n}, \tag{5}$$

where the Hessian is positive semi-definite (PSD) with one zero eigenvalue.[1]

**Related Work.** The SK algorithm has long been known to enjoy an exponential convergence rate for minimizing (4) (Franklin & Lorenz, 1989; Knight, 2008). However, at low temperatures, this fast rate does not predict non-asymptotic behavior well due to a large constant. Altschuler et al. (2017) provided a simple analysis, in which they proved a rate $\widetilde{O}(n^2 \varepsilon^{-3})$ for the SK algorithm, where $\langle P - P^*, C \rangle \leq \varepsilon$. A simple routine for rounding near-feasible plans onto $\mathcal{U}(\boldsymbol{r}, \boldsymbol{c})$ was introduced and is now widely adopted. They also proposed a new algorithm, Greenkhorn, and showed a matching complexity bound. Unlike SK, Greenkhorn scales one greedily selected row/column at a time, which limits GPU utilization unless $n$ is extremely large. The complexity bounds for Sinkhorn and

---

[1] Since $\boldsymbol{r}(P) = P\mathbf{1} = D(\boldsymbol{r}(P))\mathbf{1}$ and $\boldsymbol{c}(P) = P^\top\mathbf{1} = D(\boldsymbol{c}(P))\mathbf{1}$, we have $\nabla^2 g(\mathbf{1}, -\mathbf{1}) = \mathbf{0}$.

Greenkhorn were later improved to $\widetilde{O}(n^2\varepsilon^{-2})$ (Dvurechensky et al., 2018; Lin et al., 2019). However, our experiments suggest Sinkhorn typically behaves like $\widetilde{O}(n^2\varepsilon^{-1})$. Moreover, Kemertas et al. (2025) showed it can enjoy better performance at lower temperatures if tuned.

Dvurechensky et al. (2018) proposed an Adaptive Primal-Dual Accelerated Gradient Descent (APDAGD) algorithm for solving the dual EOT problem (4). Lin et al. (2019) provided a refined rate of $\widetilde{O}(n^{5/2}\varepsilon^{-1})$ for APDAGD and proposed a generalization APDAMD, which applied mirror descent to (4). The complexity for APDAMD was shown to be $\widetilde{O}(n^2\sqrt{c}/\varepsilon)$, where $c \in (0, n]$ is a constant. Following Altschuler et al. (2017), Lin et al. (2019) measured speed in terms of the number of row/col updates in their experiments (rather than wall-clock time) and only considered a strong regularization (low precision) setting. Targeting higher precision, Jambulapati et al. (2019) proposed an algorithm for the OT problem that is not based on entropic regularization, with complexity $\widetilde{O}(n^2\varepsilon^{-1})$. While this rate is theoretically state-of-the-art, Jambulapati et al. (2019) noted that Sinkhorn iteration, when aggressively tuned, outperforms all other methods empirically (including their own). Guminov et al. (2021) proposed an Accelerated Alternating Minimization (AAM) algorithm, combining Nesterov's momentum and Sinkhorn-type block coordinate descent, with complexity $\mathcal{O}(n^{5/2}\varepsilon^{-1})$.

While APDAMD applied mirror descent to (4), MDOT of Kemertas et al. (2025) applied it to (1) and recovered connections to temperature annealing methods (see details in Sec. 2.2), such as those of Schmitzer (2019) and Feydy (2020). For instance, Alg. 3.5. of Feydy (2020) takes a single Sinkhorn update every time the temperature is decayed; while this can compute rough approximations quickly at high temperatures, a single Sinkhorn update is insufficient for keeping the dual objective value in check, so that their approach hits a precision wall as we empirically show; see also Xie et al. (2020). Ballu & Berthet (2023) derive a similar algorithm, but they guarantee convergence by maintaining a running average of plans $P$ computed this way. While effective at low precision and easy to implement on a GPU, this algorithm exhibits $\widetilde{O}(n^2\varepsilon^{-2})$ dependence on error. Most closely related to ours is the work of Kemertas et al. (2025), as we build on MDOT. In addition to the temperature annealing framework, Kemertas et al. (2025) proposed an algorithm (PNCG) to minimize (4) at each new value of the temperature, based on a non-linear conjugate gradient method (Fletcher & Reeves, 1964). The approach introduced here has several benefits over PNCG, including added ease of theoretical analysis, faster runtime in practice and minimal line search overhead. While second order methods have been considered for OT (Mérigot, 2011; Blondel et al., 2018), they have not been implemented on GPUs with strong empirical performance in high dimensions to our knowledge. Indeed, Tang et al. (2024) also developed a 2nd order method recently, but their Hessian sparsification strategy is more amenable to a CPU setting, and as such was only tested on CPUs for $n = 784$.

## 2.2 TEMPERATURE ANNEALING AS MIRROR DESCENT

A well-known strategy to deal with the difficulty of solving (4) under weak regularization is annealing the temperature $\gamma^{-1}$ in (3) gradually towards zero. When viewed as mirror descent on (2), temperature annealing strategies (e.g., see Schmitzer (2019)) amount to a particular initialization of the dual variables $(\boldsymbol{u}, \boldsymbol{v})$ in successive instances of (4) given approximate solutions at prior $\gamma$ (Kemertas et al., 2025). Each dual problem (4) at a given $\gamma^{(t+1)}$ for $t \geq 1$ is warm-started in some neighborhood of the solution given some near-optimal $\boldsymbol{z}^{(t)} = (\boldsymbol{u}^{(t)}, \boldsymbol{v}^{(t)}) \in \mathbb{R}^{2n}$. The MDOT framework of Kemertas et al. (2025) specifically initializes $\boldsymbol{z}^{(t+1)}$ via a Taylor approximation with respect to $\gamma$ under backward finite differencing. Further, they proposed to use a more stringent tolerance $O(\gamma^{-p})$ for $\|\nabla g\|_1$ (given some $p \geq 1$), whereas prior work used $O(\gamma^{-1})$ (Altschuler et al., 2017; Lin et al., 2019). Their tuned choice $p = 1.5$ was shown to improve performance under weak regularization.

The pseudo-code for MDOT is shown in Alg. 1, where some routines are defined with "$\cdots$" as a placeholder for extra parameters that may be required by specific implementations. Given $\gamma^{(t)}$ in each iteration $t \geq 1$, the algorithm picks a tolerance $\varepsilon_{\mathrm{d}}$ for the dual gradient norm (L4). Then, marginals $\boldsymbol{r}, \boldsymbol{c}$ are smoothed in L5 for numerical stability or improved convergence by mixing in the uniform distribution (with a combined weight of at most $\varepsilon_{\mathrm{d}}/2$) to stay away from the boundary of $\mathcal{U}(\boldsymbol{r}, \boldsymbol{c})$. L6 initializes the transport plan $P$ to be the independence coupling $\tilde{\boldsymbol{r}}\tilde{\boldsymbol{c}}^{\top}$, i.e., the solution of (2) for $\gamma \to 0$ over $\mathcal{U}(\tilde{\boldsymbol{r}}, \tilde{\boldsymbol{c}})$. In L7, minimizing (4) to $\varepsilon_{\mathrm{d}}/2$ tolerance for the smoothed marginals guarantees $\|\nabla g(\boldsymbol{z}; \gamma)\|_1 \leq \varepsilon_{\mathrm{d}}$ by triangle inequality. Next, we add an AdjustSchedule routine in L8, whereas Kemertas et al. (2025) used a fixed decay rate, setting $q^{(t+1)} \leftarrow q^{(t)}$. After the temperature decay (L9), the dual variables are warm-started in L10 via an approximate 1st order expansion for the next

iteration. Finally, the plan as given by (3) is rounded onto $\mathcal{U}(\boldsymbol{r}, \boldsymbol{c})$ in L14 via Alg. 2 of Altschuler et al. (2017). Then, given that $\|\nabla g(\boldsymbol{z}; \gamma_{\mathrm{f}})\|_1 \leq \gamma_{\mathrm{f}}^{-1} \min(H(\boldsymbol{r}), H(\boldsymbol{c}))$ the user is guaranteed error $\langle P - P^*, C \rangle \leq 2\gamma_{\mathrm{f}}^{-1} \min(H(\boldsymbol{r}), H(\boldsymbol{c}))$ in the worst case (assuming $\boldsymbol{r}(P) = \boldsymbol{r}$ or $\boldsymbol{c}(P) = \boldsymbol{c}$ at loop termination). In Section 3, we develop and tune specific algorithms in tandem to carry out L5, 7 and 8 of Alg. 1 to maximize efficiency, and maintain GPU parallelization and numerical stability.

## 2.3 TRUNCATED NEWTON METHODS

For minimizing (4) in L7 of Alg. 1, we will rely on truncated Newton methods, which are briefly reviewed here (see Ch. 7 of Nocedal & Wright (2006) and Nash (2000) for a survey). While Newton's method for non-linear optimization seeks an exact solution $\boldsymbol{d}_k$ of the linear system $\nabla^2 g_k \, \boldsymbol{d}_k = -\nabla g_k$ at each optimization step $k$ (typically combined with line search or trust-region methods), truncated Newton methods find an approximate/inexact solution by "truncating" an iterative solver of the linear system (LS), e.g., a linear conjugate gradient (linear CG) algorithm. The particular termination criteria for the LS solver dictates the order of convergence;

---

**Algorithm 1** MDOT($C, \boldsymbol{r}, \boldsymbol{c}, \gamma_{\mathrm{i}}, \gamma_{\mathrm{f}}, p \geq 1, q > 1$)

1: $t \leftarrow 1$, done $\leftarrow$ false, $\gamma^{(1)} \leftarrow \gamma_{\mathrm{i}} \wedge \gamma_{\mathrm{f}}, q^{(1)} \leftarrow q, \gamma^{(0)} \leftarrow 0$
2: **while** not done **do**
3: $\quad$ done $\leftarrow \gamma^{(t)} = \gamma_{\mathrm{f}}$
4: $\quad \varepsilon_{\mathrm{d}} \leftarrow \min\left(H(\boldsymbol{r}), H(\boldsymbol{c})\right) \big/ (\gamma^{(t)})^p$
5: $\quad \tilde{\boldsymbol{r}}, \tilde{\boldsymbol{c}} \leftarrow \text{SmoothMarginals}(\boldsymbol{r}, \boldsymbol{c}, \varepsilon_{\mathrm{d}}/2; \cdots)$
6: $\quad$ **if** $t = 1$ **then** $\boldsymbol{z}^{(0)} \leftarrow (\log \tilde{\boldsymbol{r}}, \log \tilde{\boldsymbol{c}}), \boldsymbol{z}^{(1)} \leftarrow \boldsymbol{z}^{(0)}$
7: $\quad \boldsymbol{z}^{(t)} \leftarrow \text{BregmanProject}(\boldsymbol{z}^{(t)}, \gamma^{(t)}, C, \tilde{\boldsymbol{r}}, \tilde{\boldsymbol{c}}, \varepsilon_{\mathrm{d}}/2)$
8: $\quad q^{(t+1)} \leftarrow \text{AdjustSchedule}(q^{(t)}; \cdots)$
9: $\quad \gamma^{(t+1)} \leftarrow q^{(t+1)} \gamma^{(t)} \wedge \gamma_{\mathrm{f}}$
10: $\quad \boldsymbol{z}^{(t+1)} \leftarrow \boldsymbol{z}^{(t)} + \frac{\gamma^{(t+1)} - \gamma^{(t)}}{\gamma^{(t)} - \gamma^{(t-1)}}\left(\boldsymbol{z}^{(t)} - \boldsymbol{z}^{(t-1)}\right)$
11: $\quad t \leftarrow t + 1$
12: **end while**
13: $(\boldsymbol{u}, \boldsymbol{v}) \leftarrow \boldsymbol{z}^{(t-1)}, P \leftarrow \exp\{\boldsymbol{u}\mathbf{1}_n^\top + \mathbf{1}_n\boldsymbol{v}^\top - \gamma_{\mathrm{f}}C\}$
14: Output $P \leftarrow \text{Round}(P, \boldsymbol{r}, \boldsymbol{c})$

---

more stringent criteria yields higher order of convergence (up to quadratic), but requires more iterations for the LS solver (inner-most loop).

In particular, define the residual $\boldsymbol{e}_k := \nabla^2 g_k \, \boldsymbol{d}_k + \nabla g_k$ of the system in step $k$ for the *approximate Newton direction* $\boldsymbol{d}_k$. Given $\eta_k \in (0, 1)$, the LS solver is terminated when

$$\|\boldsymbol{e}_k\| \leq \eta_k \|\nabla g_k\| \tag{6}$$

for some norm. If the Hessian is continuous in some neighborhood of the solution and we start sufficiently close to the solution, we have $\|\nabla g_{k+1}\| \leq (\eta_k + o(1)) \|\nabla g_k\|$. Then, choosing *the forcing sequence* $\{\eta_k\}$ such that $\eta_k = O(\|\nabla g_k\|^{r-1})$ for $r \in (1, 2)$, one obtains Q-superlinear local convergence of order $r$ assuming $\lim_{k \to \infty} \eta_k = 0$ (Nocedal & Wright, 2006).

## 3 MIXING TRUNCATED NEWTON & SINKHORN FOR BREGMAN PROJECTIONS

This section is organized as follows. In Section 3.1, we first develop a technique for obtaining the truncated Newton direction with convergence guarantees (Alg. 2) and introduce a helper routine (Alg. 3) to improve the convergence rate of this algorithm. In Sec. 3.2, we integrate this approach with backtracking line search to arrive at a Bregman projection algorithm (L7 of Alg. 1). Its per-step cost and local convergence properties are shown theoretically. Then, in Sec. 3.3 an adaptive temperature decay schedule (L8 of Alg. 1) is proposed to maximally exploit the high rate of local convergence and its use is empirically demonstrated. Lastly, in Sec. 3.4, we discuss precautionary measures for numerical stability of this technique in practice via marginal smoothing as in L5 of Alg. 1.

## 3.1 THE DISCOUNTED HESSIAN AND THE BELLMAN EQUATIONS

Suppose we are interested in finding an approximate solution to the Newton system for the dual (Bregman projection) problem (4) with a linear CG solver. In this section, we propose a particular positive-definite (PD) approximation of the PSD Hessian in (5) for two reasons. First, although linear CG should converge in theory despite the presence of a zero eigenvalue (Axelsson, 2003), infinitesimal numerical errors along this zero-eigendirection, as well as other unknown near-zero eigendirections, may compound and lead to numerical instabilities due to machine precision limits. Second, since our

PD substitute for the Hessian allows for the use of matrix inversion, this is convenient for obtaining theoretical guarantees. This approach also yields connections to reinforcement learning that may be of separate interest. Henceforth, we define the $\rho$-discounted Hessian for $\rho \in [0, 1)$:

$$\nabla^2 g(\rho) := \begin{pmatrix} \mathbf{D}(\boldsymbol{r}(P)) & \sqrt{\rho}P \\ \sqrt{\rho}P^\top & \mathbf{D}(\boldsymbol{c}(P)) \end{pmatrix}. \tag{7}$$

By the Gerschgorin Circle Theorem, all eigenvalues $\lambda_i(\nabla^2 g(\rho)) \geq (1 - \sqrt{\rho})\min(\boldsymbol{r}(P)_{\min}, \boldsymbol{c}(P)_{\min})$ are positive, so that $\nabla^2 g(\rho)$ is invertible. The block matrix inversion formula yields:

$$\nabla^2 g(\rho)^{-1} = \begin{pmatrix} F_{\boldsymbol{r}}(\rho)^{-1} & -\sqrt{\rho}F_{\boldsymbol{r}}(\rho)^{-1}P_{\boldsymbol{c}}^\top \\ -\sqrt{\rho}P_{\boldsymbol{c}}F_{\boldsymbol{r}}(\rho)^{-1} & F_{\boldsymbol{c}}(\rho)^{-1} \end{pmatrix}, \tag{8}$$

where we used the following definitions:

$$P_{\boldsymbol{r}} := \mathbf{D}(\boldsymbol{r}(P))^{-1}P, \quad P_{\boldsymbol{c}} := \mathbf{D}(\boldsymbol{c}(P))^{-1}P^\top$$
$$P_{\boldsymbol{rc}} := P_{\boldsymbol{r}}P_{\boldsymbol{c}}, \quad P_{\boldsymbol{cr}} := P_{\boldsymbol{c}}P_{\boldsymbol{r}} \tag{9}$$
$$F_{\boldsymbol{r}}(\rho) := \mathbf{D}(\boldsymbol{r}(P))(I - \rho P_{\boldsymbol{rc}}), \quad F_{\boldsymbol{c}}(\rho) := \mathbf{D}(\boldsymbol{c}(P))(I - \rho P_{\boldsymbol{cr}}).$$

Here, $P_{\boldsymbol{r}}, P_{\boldsymbol{c}}, P_{\boldsymbol{rc}}$ and $P_{\boldsymbol{cr}}$ are irreducible row-stochastic matrices since all their entries are positive and we have $P_{\boldsymbol{r}}\mathbf{1} = \mathbf{D}(\boldsymbol{r}(P))^{-1}P\mathbf{1} = \mathbf{D}(\boldsymbol{r}(P))^{-1}\boldsymbol{r}(P) = \mathbf{1}$ (likewise for $P_{\boldsymbol{c}}$).[2] Now, multiplying on the left by (8) both sides of the *discounted Newton system* $\nabla^2 g(\rho)\,\boldsymbol{d} = -\nabla g$ and re-arranging, we arrive at the well-known Bellman equations central in reinforcement learning:

$$\boldsymbol{d_u} = \boldsymbol{s_{uv}} + \rho P_{\boldsymbol{rc}}\boldsymbol{d_u}, \qquad \boldsymbol{d_v} = \boldsymbol{s_{vu}} + \rho P_{\boldsymbol{cr}}\boldsymbol{d_v}, \tag{10}$$

where $P_{\boldsymbol{rc}}$ and $P_{\boldsymbol{cr}}$ serve as "transition matrices", and "reward vectors" are given by

$$\boldsymbol{s_{uv}} = -D(\boldsymbol{r}(P))^{-1}(\nabla_{\boldsymbol{u}}g - \sqrt{\rho}P_{\boldsymbol{c}}^\top\nabla_{\boldsymbol{v}}g), \qquad \boldsymbol{s_{vu}} = -D(\boldsymbol{c}(P))^{-1}(\nabla_{\boldsymbol{v}}g - \sqrt{\rho}P_{\boldsymbol{r}}^\top\nabla_{\boldsymbol{u}}g). \tag{11}$$

That is, the *discounted Newton direction* $\boldsymbol{d} = (\boldsymbol{d_u}, \boldsymbol{d_v})$ corresponds to the fixed point of the Bellman equations (or, "state-value function" of the finite Markov reward processes) in (10) and can be written as two $n$-variable PD linear systems rather than a single $2n$-variable PSD linear system. Further, without loss of generality, assuming that $\nabla_{\boldsymbol{v}}g = \mathbf{0}$ (or equivalently $\boldsymbol{c}(P) = \boldsymbol{c}$, for example, following a single Sinkhorn update) yields a more intuitive understanding and a practical advantage. Using the form of the inverse discounted Hessian in (8), it can be shown that in this case (10) reduces to:

$$\boldsymbol{d_u} = (\boldsymbol{r}/\boldsymbol{r}(P) - \mathbf{1}) + \rho P_{\boldsymbol{rc}}\boldsymbol{d_u}, \qquad \boldsymbol{d_v} = -\sqrt{\rho}P_{\boldsymbol{c}}\boldsymbol{d_u}, \tag{12}$$

where the second system is solved via a single matrix-vector product (effectively for free), thus reducing the problem further to a single $n$-variable PD linear system. Moreover, an intuitive interpretation of the reward vector emerges, as the system now assigns a reward $r_i/r(P)_i - 1$ to row index (state) $i$. Next, we provide a theorem on the sufficiency of solving the discounted Newton system as a proxy for the undiscounted system (see Appx. A.1.3 for a more technical version of the following).

**Theorem 3.1** (Forcing sequence under discounting). *Assuming $\boldsymbol{c} = \boldsymbol{c}(P)$ and $\boldsymbol{d_v} = -P_{\boldsymbol{c}}\boldsymbol{d_u}$, define residuals $\boldsymbol{e_u}(\rho) := F_{\boldsymbol{r}}(\rho)\boldsymbol{d_u} + \nabla_{\boldsymbol{u}}g$ (cf. (12)), and $\boldsymbol{e} := \nabla^2 g\,\boldsymbol{d} + \nabla g$ (i.e., the Newton residual). For every $\tilde{\eta} < \eta/2$, $\exists \rho_0 \in [0, 1)$ such that $\forall \rho \in [\rho_0, 1]$,*

$$\|\boldsymbol{e_u}(\rho)\|_1 \leq \tilde{\eta}\|\nabla g\|_1 \implies \|\boldsymbol{e}\|_1 \leq \eta\|\nabla g\|_1. \tag{13}$$

In other words, discounting can be adopted while still enjoying the local convergence guarantees of truncated Newton methods (discussed in Sec. 2.3) for the original problem.

To find a suitable discount factor, we propose Algorithm 2, which anneals $1 - \rho$, approximately solving a sequence of linear systems to satisfy the forcing inequality (6). Note that CG in L3 is terminated when the $L_1$ norm of the residual is below $\tilde{\eta} = \eta/4$. While the specific selection $\tilde{\eta} = \eta/4$ in L3 is only for simplicity, the decay factor 4 in L4 minimizes a theoretical upper bound on the number of operations until convergence (see proof of Thm. 3.2 in Appx. A.1.4). The following provides a rate on the overall cost of obtaining the truncated Newton direction with this approach.

**Theorem 3.2** (Convergence of Algorithm 2). *Suppose $\boldsymbol{r}(P)_{\min} \geq \varepsilon_{\mathrm{d}}/(4n)$ given $\varepsilon_{\mathrm{d}} > 0$ and each step of Alg. 2 runs diagonally-preconditioned CG initialized with $\boldsymbol{d_u}^{(0)} = \mathbf{0}$. Alg. 2 terminates in*

$$\widetilde{O}\left(n^2\sqrt{\frac{(1-\mu)\chi(\boldsymbol{r}|\boldsymbol{r}(P))}{(1-\lambda_2)\eta\|\boldsymbol{r}(P) - \boldsymbol{r}\|_1}}\log\varepsilon_{\mathrm{d}}^{-1/2}\right) \tag{14}$$

*operations, where $\lambda_2 < 1$ is the $2^{nd}$ largest eigenvalue of $P_{\boldsymbol{rc}}$ and $\mu < 1$ its smallest diagonal entry.*

---

[2]We refer the reader to Appx. A.1.3 for an intuitive description of the process represented by the stochastic matrices $P_{\boldsymbol{rc}}$ and $P_{\boldsymbol{cr}}$, and their technical properties.

The proof of this theorem uses **(i)** the $O(\sqrt{\kappa})$ convergence of linear CG (Shewchuk, 1994), where condition number $\kappa = O((1-\rho)^{-1})$ in our setting, and **(ii)** an observation from the proof of Thm. 3.1 that $(1-\rho)^{-1} = O((1-\lambda_2)^{-1})$ at termination in the worst-case. However, as we discuss in more detail and validate empirically in Appx. E, this worst-case dependence on $(1-\lambda_2)^{-1}$ can be overly pessimistic; $(1-\rho)$ can be much better behaved than the spectral gap in practice. In other words, the Newton system can be discounted more aggressively than the worst-case analysis allows, thereby mitigating possible large values of $(1-\lambda_2)^{-1}$. Furthermore, the convergence of CG can be much faster when preconditioned eigenvalues are tightly clustered on $\mathbb{R}$ (Nocedal & Wright, 2006). The rate in (14) captures this added efficiency only for the case when $P_{\boldsymbol{rc}} \approx I$ so that its smallest diagonal entry $\mu \approx 1$; for example, if plan $P$ is approximately a one-to-one mapping as in the Monge discrete matching problem (Brezis, 2018).

---

**Algorithm 2** NewtonSolve($\nabla_{\boldsymbol{u}} g, P_{\boldsymbol{rc}}, \boldsymbol{r}(P), \eta$)

1: $\rho \leftarrow 0, \boldsymbol{d_u} \leftarrow -\mathbf{D}(\boldsymbol{r}(P))^{-1}\nabla_{\boldsymbol{u}} g$
2: **while** $\|F_{\boldsymbol{r}}(1)\boldsymbol{d_u} + \nabla_{\boldsymbol{u}} g\|_1 > \eta \|\nabla g\|_1$ **do**
3: $\quad \boldsymbol{d_u} \leftarrow \text{LinearCGSolve}(F_{\boldsymbol{r}}(\rho), -\nabla_{\boldsymbol{u}} g, \eta/4)$
4: $\quad \rho \leftarrow 1 - (1-\rho)/4$
5: **end while**
6: Output $\boldsymbol{d_u}$

---

**Algorithm 3** ChiSinkhorn($\boldsymbol{u}, \boldsymbol{v}, \gamma, C, \boldsymbol{r}, \boldsymbol{c}, \boldsymbol{r}(P), \varepsilon_\chi$)

1: **while** $\chi^2(\boldsymbol{r}|\boldsymbol{r}(P)) > \varepsilon_\chi$ **do**
2: $\quad \boldsymbol{u} \leftarrow \boldsymbol{u} + \log \boldsymbol{r} - \log \boldsymbol{r}(P)$
3: $\quad \boldsymbol{v} \leftarrow \log \boldsymbol{c} - \text{LSE}_c(\boldsymbol{u}\mathbf{1}_n^\top - \gamma C)$
4: $\quad \log \boldsymbol{r}(P) \leftarrow \boldsymbol{u} + \text{LSE}_r(\mathbf{1}_n\boldsymbol{v}^\top - \gamma C)$
5: **end while**
6: Output $\boldsymbol{u}, \boldsymbol{v}, \boldsymbol{r}(P)$

---

Next, to control the possible dependence in (14) of the ratio $\chi(\boldsymbol{r}|\boldsymbol{r}(P))/\|\boldsymbol{r}(P) - \boldsymbol{r}\|_1$ on $\varepsilon_{\mathrm{d}}$, we propose to run Sinkhorn iteration before Alg. 2 is called until $\chi^2(\boldsymbol{r}|\boldsymbol{r}(P))$ is suitably bounded. Given dual variables $(\boldsymbol{u}, \boldsymbol{v})$ and current row sum $\boldsymbol{r}(P)$ where $\boldsymbol{c} = \boldsymbol{c}(P)$ by assumption, Alg. 3 performs this auxiliary task while maintaining $\boldsymbol{c} = \boldsymbol{c}(P)$.

**Lemma 3.3** (Convergence of Algorithm 3). *Assuming that $\|\boldsymbol{r}(P)/\boldsymbol{r}\|_\infty < \infty$ and $\|\boldsymbol{r}/\boldsymbol{r}(P)\|_\infty < \infty$, Algorithm 3 converges in $O(n^2/\varepsilon_\chi)$ operations.*

## 3.2 PROJECTING ONTO THE FEASIBLE POLYTOPE

Combining Algorithms 2 and 3 with backtracking line search, we arrive at the TruncatedNewton-Project algorithm shown in Alg. 4 for solving (4). The choice of $\eta$ in L5 and the requirement that $\chi^2(\boldsymbol{r}|\boldsymbol{r}(P)) \leq \varepsilon_{\mathrm{d}}^{2/5}$ by choosing $\varepsilon_\chi = \varepsilon_{\mathrm{d}}^{2/5}$ in L4, to control the ratio in (14), together yield the following corollary of Thm. 3.2 and Lemma 3.3 (see proof in Appx. A.1.5).

**Corollary 3.4** (Per-step Cost of Algorithm 4). *If the backtracking line search in Alg. 4 converges in $S$ iterations, then an iteration of Alg. 4 costs $\widetilde{O}(n^2(S + \varepsilon_{\mathrm{d}}^{-2/5}(1-\lambda_2)^{-1/2}))$ operations, where $\lambda_2 < 1$ is the 2nd largest eigenvalue of $P_{\boldsymbol{rc}}$ defined as in (9) and evaluated at $\boldsymbol{u}, \boldsymbol{v}$ (cf. (3)).*

In the next section, we outline an adaptive temperature annealing strategy to minimize the added cost of line search by initializing close to the solution. First, we pause for the next theorem, showing that Alg. 4 enjoys local quadratic convergence if $\eta = O(\|\nabla g\|_1)$ as in L5.

**Theorem 3.5** (Per-step Improvement of Algorithm 4). *Given a descent direction $\boldsymbol{d} = (\boldsymbol{d_u}, -P_{\boldsymbol{c}}\boldsymbol{d_u})$ such that $\|\boldsymbol{e}\|_1 = \|\nabla^2 g_k \boldsymbol{d} + \nabla g_k\|_1 \leq \eta \|\nabla g_k\|_1$, let $\alpha \in (0,1]$ be the step size found via backtracking line search in the $k^{\mathrm{th}}$ step of Alg. 4. Then, $\nabla g_{k+1} \coloneqq \nabla g(\boldsymbol{u} + \alpha\boldsymbol{d_u}, \boldsymbol{v} - \alpha P_{\boldsymbol{c}}\boldsymbol{d_u})$ satisfies*

$$\|\nabla g_{k+1}\|_1 \leq (1 - \alpha + \alpha\eta)\|\nabla g_k\|_1 + \alpha\sqrt{\alpha}\, O(\|\nabla g_k\|_1^2). \tag{15}$$

The result differs from typical quadratic convergence results in two important ways: **(i)** we did not assume a Lipschitz Hessian, but instead leveraged the Armijo condition and the specific form of our descent direction, and **(ii)** we bounded the $L_1$ norm of the gradient (rather than $L_2$), which is more commonly used in optimal transport algorithms as stopping criteria. Given (15), we select $\eta$ in L5 of Alg. 4 with the local quadratic rate in mind, but avoid over-solving the system when $\|\nabla g\|_1$ is already close to the target $\varepsilon_{\mathrm{d}}$ by taking $\eta$ to be the maximum of $\|\nabla g\|_1$ and $0.8\varepsilon_{\mathrm{d}}/\|\nabla g\|_1$.

## 3.3 INITIALIZING NEAR THE SOLUTION VIA ADAPTIVE MIRROR DESCENT

In this section, we outline a practical strategy for initializing the dual problem (4) sufficiently near the solution so that **(i)** the cost of line search is minimized ($\alpha = 1$ is almost always admissible),

---

**Algorithm 4** TruncatedNewtonProject($\boldsymbol{u}, \boldsymbol{v}, \gamma, C, \boldsymbol{r}, \boldsymbol{c}, \varepsilon_{\mathrm{d}}$)

---

1:  $\boldsymbol{v} \leftarrow \log \boldsymbol{c} - \mathrm{LSE}_c(\boldsymbol{u}\mathbf{1}_n^\top - \gamma C)$                              ▷ Ensure $\boldsymbol{c}(P) = \boldsymbol{c}$.
2:  $\log \boldsymbol{r}(P) \leftarrow \boldsymbol{u} + \mathrm{LSE}_r(\mathbf{1}_n\boldsymbol{v}^\top - \gamma C)$
3:  **while** $\|\nabla g\|_1 = \|\boldsymbol{r} - \boldsymbol{r}(P)\|_1 > \varepsilon_{\mathrm{d}}$ **do**
4:     $\boldsymbol{u}, \boldsymbol{v}, \boldsymbol{r}(P) \leftarrow \mathrm{ChiSinkhorn}(\boldsymbol{u}, \boldsymbol{v}, \gamma, C, \boldsymbol{r}, \boldsymbol{c}, \boldsymbol{r}(P), \varepsilon_{\mathrm{d}}^{2/5})$       ▷ Choosing $\varepsilon_\chi = \varepsilon_{\mathrm{d}}^{2/5}$.
5:     $\eta \leftarrow \|\nabla g\|_1 \vee 0.8\varepsilon_{\mathrm{d}}/\|\nabla g\|_1$              ▷ Expect $\|\nabla g_{k+1}\|_1 \leq \eta \|\nabla g_k\|_1$.
6:     $\boldsymbol{d_u} \leftarrow \mathrm{NewtonSolve}(\nabla_{\boldsymbol{u}} g, P_{\boldsymbol{rc}}, \boldsymbol{r}(P), \eta)$                  ▷ See Algorithm 2.
7:     $\boldsymbol{d_v} \leftarrow -P_c\boldsymbol{d_u}$                                     ▷ As in (12).
8:     $\alpha \leftarrow 1$                                         ▷ Initial guess for step size.
9:     $\log \boldsymbol{c}(P) \leftarrow \boldsymbol{v} + \alpha\boldsymbol{d_v} + \mathrm{LSE}_c((\boldsymbol{u} + \alpha\boldsymbol{d_u})\mathbf{1}_n^\top - \gamma C)$
10:    **while** $\|\boldsymbol{c}(P)\|_1 - 1 > 0.99\alpha\langle -\nabla_{\boldsymbol{u}} g, \boldsymbol{d_u}\rangle$ **do**      ▷ *Armijo condition*. See Appx. A.1.6.
11:       $\alpha \leftarrow 0.5\alpha$                               ▷ Backtracking line search.
12:       $\log \boldsymbol{c}(P) \leftarrow \boldsymbol{v} + \alpha\boldsymbol{d_v} + \mathrm{LSE}_c((\boldsymbol{u} + \alpha\boldsymbol{d_u})\mathbf{1}_n^\top - \gamma C)$
13:    **end while**
14:    $\boldsymbol{u} \leftarrow \boldsymbol{u} + \alpha\boldsymbol{d_u}, \boldsymbol{v} \leftarrow \boldsymbol{v} + \alpha\boldsymbol{d_v}$
15:    $\boldsymbol{v} \leftarrow \boldsymbol{v} + \log \boldsymbol{c} - \log \boldsymbol{c}(P)$                          ▷ Ensure $\boldsymbol{c}(P) = \boldsymbol{c}$.
16:    $\log \boldsymbol{r}(P) \leftarrow \boldsymbol{u} + \mathrm{LSE}_r(\mathbf{1}_n\boldsymbol{v}^\top - \gamma C)$
17: **end while**
18: $\boldsymbol{u} \leftarrow \boldsymbol{u} + \log \boldsymbol{r} - \log \boldsymbol{r}(P)$       ▷ Since $\log \boldsymbol{r}(P)$ is readily available, take a Sinkhorn step.
19: Output $\boldsymbol{u}, \boldsymbol{v}$

---

and **(ii)** the last term in (15) is negligible. In Table 1, we observe that with a sufficiently slow fixed temperature decay schedule (i.e., setting $q^{(t+1)} \leftarrow q^{(t)}$ in L8 of Alg. 1), the extra cost of line search (as well as ChiSinkhorn) disappears almost entirely (given small enough $\gamma^{(1)}$). However, with too slow temperature decay schedules ($q$ too close to 1), the overhead due to relatively costly LSE reductions (in lines 1, 2, 9 and 16 of Alg. 4) may slow down the overall algorithm (see $q = 2^{1/8}$ in Table 1). To eliminate the need for tuning $q$, we develop an update rule to adjust the schedule as in L8 of Alg. 1; we seek to minimize the number of mirror descent steps while staying in the superlinear convergence zone for consecutive instances of problem (4). To this end, we first define the following parameter of interest, which is the ratio of the *actual reduction* in gradient norm to the *predicted reduction* given by (15) for $\alpha = 1$ (in the ideal case, dropping the last term) at step $k$ of Alg. 4:

$$\delta_k := \frac{\|\nabla g_k\|_1 - \|\nabla g_{k+1}\|_1}{(1 - \eta_k)\|\nabla g_k\|_1}. \tag{16}$$

This formula for $\delta_k$ is inspired by trust-region methods, which update the trust-region size based on the ratio of actual to predicted decrease under a model of the objective (rather than the gradient norm as we use here) (Nocedal & Wright, 2006). Let $\delta_{\min}^{(t)}$ be the smallest $\delta_k$ among the iterates of Alg. 4 at step $t$ of Alg. 1. In L5 of Alg. 1, we heuristically set

$$q^{(t+1)} \leftarrow 2 \wedge (q^{(t)})^2 \qquad \text{if} \quad \delta_{\min}^{(t)} > 0.95$$
$$q^{(t+1)} \leftarrow \sqrt{q^{(t)}} \qquad\quad \text{if} \quad \delta_{\min}^{(t)} < 0.8$$
$$q^{(t+1)} \leftarrow q^{(t)} \qquad\qquad\;\; \text{otherwise.}$$

| Subroutine     $q$ | $2^{1/8}$ | $2^{1/4}$ | $2^{1/2}$ | $2^1$ | $2^2$ | Adaptive |
|---|---|---|---|---|---|---|
| NewtonSolve | 1870 | 1725 | 1744 | 2223 | 3281 | 1831 |
| Line Search | 8 | 8 | 8 | 48 | 226 | 24 |
| ChiSinkhorn | 8 | 8 | 8 | 8 | 96 | 8 |
| Mirror Descent | 105 | 53 | 27 | 14 | 8 | 23 |
| Total (operations) | 4623 | 3315 | 2760 | 3178 | 4369 | 2795 |
| Total (seconds) | 2.84 | 1.99 | 1.56 | 1.76 | 2.41 | 1.55 |

Table 1: Breakdown of number of $O(n^2)$ operations per subroutine and total wall-clock time for the upsampled MNIST dataset ($n = 4096$) with $L_1$ distance cost. Alg. 1 is called with $\gamma_{\mathrm{i}} = 2^5, \gamma_{\mathrm{f}} = 2^{18}$ and $p = 1.5$. For the adaptive approach, $q = 2$ initially. Results show median over 100 problems.

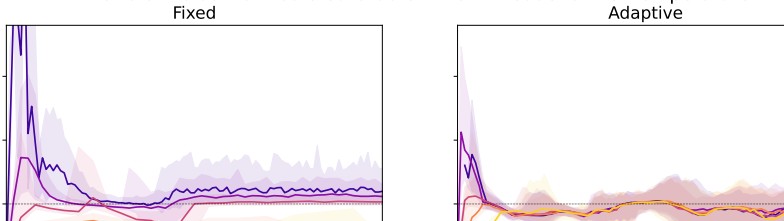

Figure 1: Ratio $\delta_k$ of actual to theoretically predicted reduction in $\|\nabla g_k\|_1$ per step for fixed **(left)** and adaptive **(right)** temperature decay (initialized with $q^{(1)} = q$). Each $\delta_k$ is the median at iteration $t$ of Alg. 1. Shaded areas show 80% confidence intervals around median over 100 random problems from the upsampled MNIST dataset ($n = 4096$) with normalized $L_1$ distance cost ($\max_{i,j} |C_{i,j}| = 1$).

In Fig. 1 (left), we show that with a sufficiently slow fixed schedule, $\delta_k$ typically remains near 1 throughout the execution of Alg. 1. Fig. 1 (right) further confirms that the adaptive update rule arrives at such a schedule regardless of the initial value of $q^{(1)}$, thereby showing its utility in eliminating the need for tuning. Table 1 verifies that the schedules found by the adaptive updating perform similarly to the best fixed schedule across OT problems. Section 4 details the experimental setup here, and further adds benchmarking against a suite of alternative algorithms in the literature.

### 3.4 ASYMMETRIC MARGINAL SMOOTHING FOR NUMERICAL STABILITY

Recall from (9) the form of the coefficient matrix $F_r(\rho) = D(\boldsymbol{r}(P))(I - \rho P_{\boldsymbol{rc}})$ of the $n$-variable PD linear system that we are interested in solving. Since the smallest diagonal entry of $F_r(\rho)$ can be almost as small as $(1 - \rho)\boldsymbol{r}(P)_{\min}$ in the worst case, diagonal preconditioning used in Alg. 2 may cause numerical instabilities when solving the linear system (due to infinitesimal entries in $\boldsymbol{r}(P)$ for $\rho \approx 1$). To this end we require that, after

| Total cost ╲ $w_r$ | 0.25 | 0.35 | 0.45 |
|---|---|---|---|
| Median (ops.) | 3431 | 2750 | 2795 |
| 90th %ile (ops.) | 7622 | 3852 | 3869 |
| Median (sec.) | 1.96 | 1.56 | 1.55 |
| 90th %ile (sec.) | 4.41 | 2.16 | 2.13 |

Table 2: Comparison of median and 90th percentile performance for varying smoothing weight $w_r$.

smoothing, $\tilde{r}_{\min}$ is bounded away from zero, and find that running Alg. 3 in advance to bound $\chi^2(\tilde{\boldsymbol{r}}|\boldsymbol{r}(P))$ (i.e., the variance of $\tilde{\boldsymbol{r}}/\boldsymbol{r}(P) - \boldsymbol{1}$) is sufficient for stable behavior in practice. Since log-domain Sinkhorn updates to ensure $\boldsymbol{c}(P) = \tilde{\boldsymbol{c}}$ are numerically stable (Feydy, 2020), we allocate our "smoothing budget" $\varepsilon_d/2$ mostly for the row-marginal $\boldsymbol{r}$. Specifically, in L5 of Alg. 1, we set:

$$\tilde{\boldsymbol{r}} \leftarrow (1 - w_{\boldsymbol{r}}\varepsilon_d)\boldsymbol{r} + (w_{\boldsymbol{r}}\varepsilon_d/n)\boldsymbol{1}, \qquad \tilde{\boldsymbol{c}} \leftarrow (1 - w_{\boldsymbol{c}}\varepsilon_d)\boldsymbol{c} + (w_{\boldsymbol{c}}\varepsilon_d/n)\boldsymbol{1},$$

where $w_{\boldsymbol{r}} + w_{\boldsymbol{c}} = 1/2$ and $w_{\boldsymbol{r}} > w_{\boldsymbol{c}}$. This is in contrast to the more standard symmetric smoothing $w_{\boldsymbol{r}} = w_{\boldsymbol{c}}$ (Dvurechensky et al., 2018; Lin et al., 2019). We repeat the experiments in Table 1, but this time ablating $w_{\boldsymbol{r}}$ (using the adaptive schedule introduced in Sec. 3.3); Table 2 confirms empirically that asymmetric smoothing improves both stability and median performance of the overall algorithm.

## 4 EXPERIMENTS

In this section, we first provide the details of the MNIST experiments in Fig. 1 and Tables 1-2. Then we describe a color transfer problem set used for wall-clock time benchmarking of the combined MDOT-TruncatedNewton algorithm. Our setup and baseline implementations follow Kemertas et al. (2025). All experiments were run on an NVIDIA GeForce RTX 2080 Ti GPU with 64-bit precision. Appx. D provides benchmarking on 10 additional datasets from Schrieber et al. (2017), showing similar results with confidence intervals and including operation counts alongside wall-clock time.

### 4.1 EXPERIMENTAL SETUP

**Upsampled MNIST.** Each image is a probability distribution over pixels, with probabilities given by $L_1$-normalized intensity values. The cost matrix $C$ is fixed across OT problems and constructed from

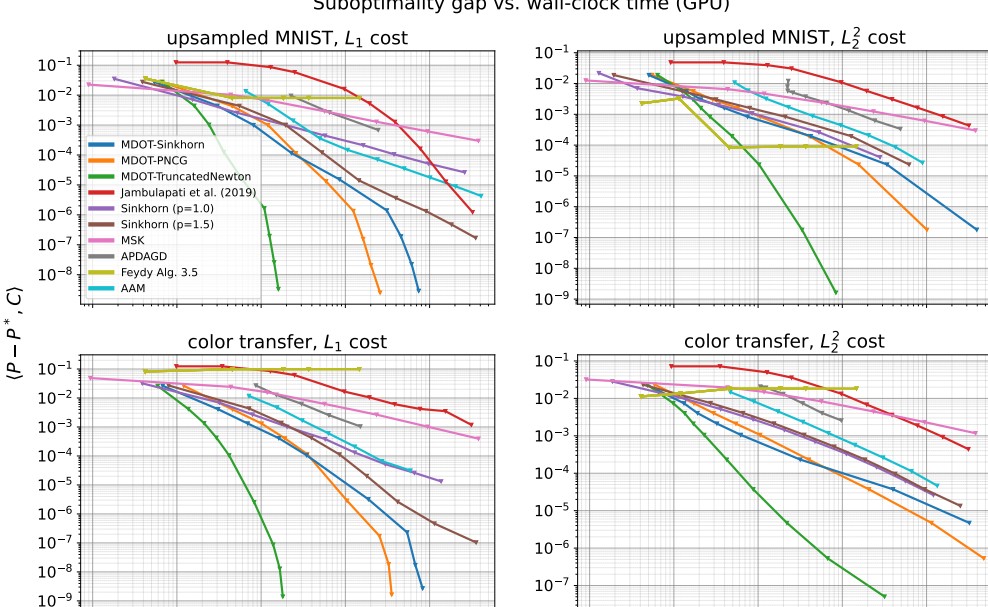

Figure 2: Error vs. wall-clock time for various algorithms. Each marker shows the optimality gap and time taken (median across 18 problems) until termination at a given hyperparameter setting, followed by rounding of the output onto $\mathcal{U}(r, c)$ via Alg. 2 of Altschuler et al. (2017). Upsampled MNIST **(top)** and color transfer **(bottom)** problem sets ($n = 4096$) using $L_1$ **(left)** and $L_2^2$ **(right)** distance costs. MDOT–TruncatedNewton outperforms others by orders of magnitude at high precision and exhibits much better practical dependence on error than best known theoretical rates $\widetilde{O}(n^2 \varepsilon^{-1})$.

pairwise $L_1$ and $L_2^2$ distances between pixel locations in 2D space. Scalar division by the max. entry ensures $\max_{ij} |C_{ij}| = 1$. MNIST images are upsampled to $64 \times 64$ resolution for benchmarking on higher dimensional problems ($n = 4096$). In contrast, Tang et al. (2024) benchmarked their CPU-based algorithm on original MNIST images ($n = 784$). Luo et al. (2023) ran their PDASMD algorithm on *downsampled* MNIST images for $n = 100$ at most; their code also runs on a CPU and includes an inner for loop of $n$ iterations which limits parallelization.

**Color Transfer.** We define the color transfer problem with all marginals set to the uniform distribution over $\Delta_n$. The cost matrix $C$ varies across problems and is constructed from pairwise $L_1$ and $L_2^2$ distances between RGB values in 3D space. Scalar division by the max. entry ensures $\max_{ij} |C_{ij}| = 1$. We use the 20 images provided by Kemertas et al. (2025), which were generated by prompting DALL-E 2 to produce vibrant, colorful images with intricate details. These are downsampled to $64 \times 64$ resolution for benchmarking on $n = 4096$ problems, except in the case of Fig 3, where the scalability of our algorithm is visually demonstrated on the original $1024 \times 1024$ images ($n \approx 10^6$) on an individual sample problem.

## 4.2 WALL-CLOCK TIME BENCHMARKING

While theoretical analysis and computational complexity provide valuable insights, it is the practical performance, measured in wall-clock time, that often determines the viability and adoption of an algorithm. In this section, we present wall-clock time benchmarking of the proposed algorithm against a broad range of available alternatives. Here, we compare to Alg. 3.5 of Feydy (2020), the Mirror Sinkhorn (MSK) algorithm of Ballu & Berthet (2023), Sinkhorn iteration with typical and stringent tolerance settings ($p = 1$ and $p = 1.5$ resp. for MDOT–Sinkhorn called with $\gamma_i = \gamma_f$), the APDAGD algorithm of Dvurechensky et al. (2018), the Mirror Prox Sherman Optimized algorithm of Jambulapati et al. (2019) and the AAM algorithm of Guminov et al. (2021). We omit comparison to APDAMD (Lin et al., 2019), APDRCD (Guo et al., 2020) and PDASMD (Luo et al., 2023), as they exhibit significantly longer convergence times in our high-dimensional, GPU-parallel setting. See Appx. E of Kemertas et al. (2025) for implementation details. In Appx. C, we also perform experiments with varying problem size $n$ and show that the dependence is no worse than $O(n^2)$ for the problems considered.

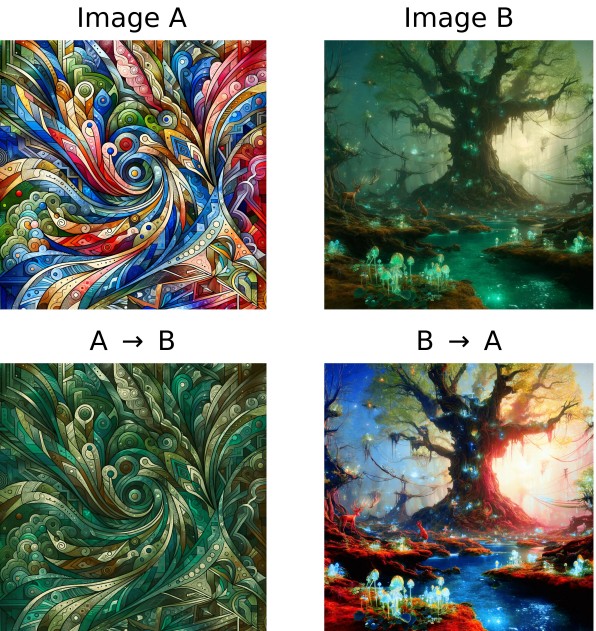

Figure 3: The MDOT–TruncatedNewton algorithm applied to a large-scale color transfer problem on $1024 \times 1024$ images ($n = 2^{20}$). For this visualization, the cost matrix is given by the $L_2^2$ distance in RGB color space, normalized so that $\max_{ij} |C_{ij}| = 1$. Final temperature is $1/\gamma_{\mathrm{f}} = 2^{-10}$. Source images (**top row**) were generated with DALL-E 2. This figure is best viewed digitally.

In Fig. 2, we observe that on a 2018-era GPU, MDOT–TruncatedNewton typically solves $n = 4096$ dimensional problems in 1-5 seconds to 6-decimal precision, demonstrating numerical stability up to 9-decimal precision across a range of realistic OT problems. In the highest precision range, it is more than $10\times$ faster than the best alternative, MDOT–PNCG of Kemertas et al. (2025). We also implement a scalable $O(n)$ memory footprint version of MDOT-TruncatedNewton, which computes via the PyKeOps package of Charlier et al. (2021) the cost matrix $C$ and the matrix $P$ on-the-fly every-time they are used in Alg. 4. As evidenced by Fig. 3, this implementation can solve very high dimensional OT problems ($n \approx 10^6$) to high precision. It leaves a memory footprint of just $\approx 600$ MBs, but takes $\approx 10$ hours. Regardless, we believe that this is an important step towards high-precision discrete OT in very high dimensions.

## 5 Conclusion

In this work, we set out to design a modular, practical algorithm to exploit the superlinear convergence of truncated Newton methods in weakly-regularized EOT. To improve the conditioning of the dual Hessian, rather than amplifying its diagonal entries as in Tikhonov regularization, we dampened off-diagonal entries (discounting) with inspiration from reinforcement learning. Then, Alg. 2 was presented for approximately solving the modified Newton system. This method of Hessian modification enabled a superlinear local convergence rate in terms of the $L_1$ norm of the gradient for the custom truncated Newton routine (Alg. 4) that used Alg. 2. We additionally introduced precautionary measures to improve the numerical stability of Alg. 4, which is crucial for reaching high precision. Lastly, Alg. 4 was integrated into a temperature annealing framework, MDOT (Kemertas et al., 2025), where adaptive temperature updates ensured superlinear convergence is maintained, a hyperparameter ($q$ in Alg. 1) is eliminated and line search overhead is minimized in practice. We implemented the resulting algorithm on a GPU and showed that it outperforms many recent algorithms by orders of magnitude in $n = 4096$ dimensions, exhibiting fast empirical rates ranging from $O(n^2 \varepsilon^{-1/6})$ to $O(n^2 \varepsilon^{-1/2})$. Furthermore, as visualized in Fig. 3, the algorithm holds potential to effortlessly scale to much larger problems ($n \approx 10^6$).

One avenue for future research is the development of a variant of Alg. 4 with a global convergence rate, which may yield an explicit rate in terms of the error $\varepsilon$ for the overall algorithm. Another direction could be a stochastic generalization of Alg. 4 that leaves an $O(nm)$ memory footprint, where $m \in [1, n]$ is a user-defined parameter given hardware constraints. This added flexibility may enable a better trade-off between runtime and memory footprint in very high dimensions ($n \gg 1000$).

ACKNOWLEDGMENTS

AMF acknowledges the support of the Natural Sciences and Engineering Research Council of Canada (NSERC) through the Discovery Grant program (2021-03701). MK acknowledges the support of NSERC through the CGS-D scholarship. Resources used in preparing this research were provided, in part, by the Province of Ontario, the Government of Canada through CIFAR, and companies sponsoring the Vector Institute.

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

# A APPENDIX

## A.1 PROOFS OF THEORETICAL RESULTS

### A.1.1 ADDITIONAL NOTATION

Here, we describe additional notation used in the proofs that follow. Given a column vector $\boldsymbol{x} \in \mathbb{R}^n$ and an $n \times n$ square matrix $A$, $\|\boldsymbol{x}\|_A^2 = \boldsymbol{x}^\top A \boldsymbol{x}$. The operator norm of a matrix is denoted $\|A\|_{p,q} = \sup_{\boldsymbol{x} \in \mathbb{R}^n} \|A\boldsymbol{x}\|_q / \|\boldsymbol{x}\|_p$. To mean $\|A\|_{p,p}$, we use the notation $\|A\|_p$. For an element-wise norm, we write $\|\mathbf{vec}(A)\|_p$, where $\mathbf{vec}(A)$ is the vectorization of the matrix $A$. A vector formed by diagonal entries of a matrix $P$ is denoted $\mathrm{diag}(P)$.

### A.1.2 DERIVATION OF THE EOT DUAL (4)

In this section, we provide the derivations for (3-4). Recall the EOT primal problem given by (2):

$$\underset{P \in \mathcal{U}(\boldsymbol{r}, \boldsymbol{c})}{\text{minimize}} \quad \langle P, C \rangle - \frac{1}{\gamma} H(P).$$

Observe that we can replace negative Shannon entropy $\langle P, \log P \rangle$ in (2) with the KL divergence (written without assuming $P \in \Delta_{n \times n}$) to the uniform distribution $U = \frac{1}{n^2} \mathbf{1}_{n \times n}$:

$$D_{\mathrm{KL}}(P|U) = \langle P, \log P \rangle - \langle P, \log \frac{1}{n^2} \mathbf{1} \rangle + 1 - \sum_{ij} P_{ij}$$

$$= \langle P, \log P \rangle - \sum_{ij} P_{ij} + 2 \log n + 1.$$

That is, replacing the negative Shannon entropy term with the above in the primal objective only increases the objective by a constant $2 \log n$ on the feasible set, since $\sum_{ij} P_{ij} = 1$ for all $P \in \mathcal{U}(\boldsymbol{r}, \boldsymbol{c})$. For convenience, we drop the $2 \log n$ term and take the primal problem given by

$$\underset{P \in \mathcal{U}(\boldsymbol{r}, \boldsymbol{c})}{\text{minimize}} \quad \langle P, C \rangle + \frac{1}{\gamma} \left( \langle P, \log P \rangle - \sum_{ij} P_{ij} + 1 \right).$$

To derive the dual problem, we write the (scaled) Lagrangian with dual variables $\boldsymbol{u}, \boldsymbol{v} \in \mathbb{R}^n$ corresponding to equality constraints $\boldsymbol{r}(P) = \boldsymbol{r}$ and $\boldsymbol{c}(P) = \boldsymbol{c}$:

$$\mathcal{L}(P, \boldsymbol{u}, \boldsymbol{v}) = \gamma \langle P, C \rangle + \langle P, \log P \rangle - \sum_{ij} P_{ij} + 1 + \langle \boldsymbol{u}, \boldsymbol{r} - \boldsymbol{r}(P) \rangle + \langle \boldsymbol{v}, \boldsymbol{c} - \boldsymbol{c}(P) \rangle. \tag{17}$$

Taking the first derivative with respect to the $ij^{\text{th}}$ entry $P_{ij}$ of $P$:

$$\frac{\partial \mathcal{L}}{\partial P_{ij}} = \gamma C_{ij} + 1 + \log P_{ij} - 1 - u_i - v_j$$

$$= \gamma C_{ij} + \log P_{ij} - u_i - v_j.$$

Setting the partial to 0:

$$\frac{\partial \mathcal{L}}{\partial P_{ij}} = 0 \iff \log P_{ij} = u_i + v_j - \gamma C_{ij}$$

$$\iff P_{ij} = \exp\{u_i + v_j - \gamma C_{ij}\},$$

where the last equation is the same as (3). Now, plugging the above $P$ into the Lagrangian (17):

$$-g(\boldsymbol{u}, \boldsymbol{v}) := \gamma \langle P, C \rangle + \langle P, \boldsymbol{u}\mathbf{1}^\top + \mathbf{1}\boldsymbol{v}^\top - \gamma C \rangle - \sum_{ij} P_{ij} + 1 + \langle \boldsymbol{u}, \boldsymbol{r} - \boldsymbol{r}(P) \rangle + \langle \boldsymbol{v}, \boldsymbol{c} - \boldsymbol{c}(P) \rangle$$

$$= \langle \boldsymbol{u}, P\mathbf{1} \rangle + \langle \boldsymbol{v}, P^\top \mathbf{1} \rangle - \sum_{ij} P_{ij} + 1 + \langle \boldsymbol{u}, \boldsymbol{r} - \boldsymbol{r}(P) \rangle + \langle \boldsymbol{v}, \boldsymbol{c} - \boldsymbol{c}(P) \rangle$$

$$= 1 - \sum_{ij} P(\boldsymbol{u}, \boldsymbol{v})_{ij} + \langle \boldsymbol{u}, \boldsymbol{r} \rangle + \langle \boldsymbol{v}, \boldsymbol{c} \rangle.$$

Maximizing $-g(\boldsymbol{u}, \boldsymbol{v})$ with respect to the dual variables, we obtain the dual problem (4):

$$\operatorname*{minimize}_{\boldsymbol{u}, \boldsymbol{v} \in \mathbb{R}^n} \quad g(\boldsymbol{u}, \boldsymbol{v}) = \sum_{ij} P(\boldsymbol{u}, \boldsymbol{v})_{ij} - 1 - \langle \boldsymbol{u}, \boldsymbol{r} \rangle - \langle \boldsymbol{v}, \boldsymbol{c} \rangle,$$

where we kept the constant $-1$ as a convention.

As an aside, note that if one assumes instead an $L_1$ normalized form for $P$ via the softmax function (given that the feasible set $\mathcal{U}(\boldsymbol{r}, \boldsymbol{c})$ is a subset of the simplex):

$$P(\boldsymbol{u}, \boldsymbol{v})_{ij} = \frac{\exp\{u_i + v_j - \gamma C_{ij}\}}{\sum_{k,l} \exp\{u_k + v_l - \gamma C_{ij}\}},$$

one obtains an alternative form for the dual objective by plugging the above into the Lagrangian (17):

$$\tilde{g}(\boldsymbol{u}, \boldsymbol{v}) = \log \sum_{ij} \exp\{u_i + v_j - \gamma C_{ij}\} - \langle \boldsymbol{u}, \boldsymbol{r} \rangle - \langle \boldsymbol{v}, \boldsymbol{c} \rangle.$$

Both the sum-of-exponents and log-sum-of-exponents forms of the dual appear in the literature; see for instance Altschuler et al. (2017); Dvurechensky et al. (2018); Lin et al. (2019) for the former and Lin et al. (2022) for the latter. Both objectives $g$ and $\tilde{g}$ have the same value whenever $P(\boldsymbol{u}, \boldsymbol{v})$ as defined in (3) is on the simplex, since $x - 1 = \log x = 0$ for $x = 1$, which is why the constant $-1$ in (4) was kept as a convention.

### A.1.3 Proof of Thm. 3.1

We start this section with an intuitive example describing the role of the row-stochastic matrix $P_{\boldsymbol{rc}}$ that was defined in (9). The discussion carries over to $P_{\boldsymbol{cr}}$ by symmetry. Next, we will list some mathematical properties that will be useful in the proofs that follow.

Suppose $\boldsymbol{r}$ and $\boldsymbol{c}$ are disjointly supported on two sets of particles $\boldsymbol{x}_1, \cdots, \boldsymbol{x}_{n_1}$ and $\boldsymbol{y}_{n_1+1}, \cdots, \boldsymbol{y}_n$ respectively, and let $n_2 := n - n_1$. That is, an $n_1 \times n_2$ transport plan $P \in \mathcal{U}(\boldsymbol{r}, \boldsymbol{c})$ maps distributions over $\boldsymbol{x}$ and $\boldsymbol{y}$ particles. Recall the definition of $P_{\boldsymbol{rc}} = P_{\boldsymbol{r}} P_{\boldsymbol{c}}$ in (9) as a product of $P_{\boldsymbol{r}} = \mathbf{D}(\boldsymbol{r})^{-1} P \in \mathbb{R}_{>0}^{n_1 \times n_2}$ and $P_{\boldsymbol{c}} = \mathbf{D}(\boldsymbol{c})^{-1} P^\top \in \mathbb{R}_{>0}^{n_2 \times n_1}$. Given some initial distribution $\boldsymbol{q} \in \Delta_{n_1}$ over $\boldsymbol{x}$-particles, $P_{\boldsymbol{r}}^\top \boldsymbol{q} \in \Delta_{n_2}$ is a distribution over $\boldsymbol{y}$-particles after transportation according to $P$. Indeed, we can show easily that $P_{\boldsymbol{r}}^\top \boldsymbol{q}$ is on the simplex:

$$\mathbf{1}^\top P_{\boldsymbol{r}}^\top \boldsymbol{q} = \mathbf{1}^\top P^\top \mathbf{D}(\boldsymbol{r})^{-1} \boldsymbol{q} = \boldsymbol{r}^\top \mathbf{D}(\boldsymbol{r})^{-1} \boldsymbol{q} = \mathbf{1}^\top \boldsymbol{q} = 1.$$

Similarly, $P_{\boldsymbol{c}}^\top P_{\boldsymbol{r}}^\top q = P_{\boldsymbol{rc}}^\top q \in \Delta_{n_1}$ is again a distribution over $\boldsymbol{x}$-particles after transporting back according to $P$. This is the stochastic process represented by $P_{\boldsymbol{rc}}$. Given any initial distribution $\boldsymbol{q}$, the process converges to the row-marginal $\boldsymbol{r}$ of $P$ if $P_{\boldsymbol{rc}}$ is applied repeatedly (as the next lemma shows). The second largest eigenvalue $\lambda_2$ of $P_{\boldsymbol{rc}}$ determines how quickly this convergence occurs (see Lemma A.3 below).

Now, we list useful technical properties of $P_{\boldsymbol{rc}}$ in our setting. Analogous claims hold for the stochastic matrix $P_{\boldsymbol{cr}}$ and the column sum $\boldsymbol{c}(P)$ by symmetry, but are omitted for brevity.

**Lemma A.1** (Properties of $P_{\boldsymbol{rc}}$). *Given a matrix $P \in \mathbb{R}^{n \times n}$ with strictly positive finite entries define $P_{\boldsymbol{rc}} = P_{\boldsymbol{r}} P_{\boldsymbol{c}} = \mathbf{D}(\boldsymbol{r}(P))^{-1} P \mathbf{D}(\boldsymbol{c}(P))^{-1} P^\top$. The following are true:*

1. *$P_{\boldsymbol{rc}}$ is an irreducible row-stochastic matrix. Its second largest eigenvalue $\lambda_2$ is strictly less than one.*

2. *The stationary distribution of $P_{\boldsymbol{rc}}$ is $\boldsymbol{r}(P)$.*

3. *$P_{\boldsymbol{rc}}$ is reversible in the sense that $\mathbf{D}(\boldsymbol{r}(P)) P_{\boldsymbol{rc}} = P_{\boldsymbol{rc}}^\top \mathbf{D}(\boldsymbol{r}(P))$, which implies all eigenvalues of $P_{\boldsymbol{rc}}$ are real.*

4. *$P_{\boldsymbol{rc}} \mathbf{D}(\boldsymbol{r}(P))^{-1} = \mathbf{D}(\boldsymbol{r}(P))^{-1} P_{\boldsymbol{rc}}^\top$.*

5. *Given some $\rho \in [0, 1)$, we have $(I - \rho P_{\boldsymbol{rc}})^{-1} \mathbf{D}(\boldsymbol{r}(P))^{-1} = \mathbf{D}(\boldsymbol{r}(P))^{-1} (I - \rho P_{\boldsymbol{rc}}^\top)^{-1}$.*

*Proof.* We prove each claim in order.

1. The vector of ones is a right-eigenvector of $P_{rc}$ with eigenvalue 1:

$$\begin{aligned}
P_{rc}\mathbf{1} &= \mathbf{D}(\boldsymbol{r}(P))^{-1}P\mathbf{D}(\boldsymbol{c}(P))^{-1}P^\top\mathbf{1} \\
&= \mathbf{D}(\boldsymbol{r}(P))^{-1}P\mathbf{D}(\boldsymbol{c}(P))^{-1}\boldsymbol{c}(P) \\
&= \mathbf{D}(\boldsymbol{r}(P))^{-1}P\mathbf{1} \\
&= \mathbf{D}(\boldsymbol{r}(P))^{-1}\boldsymbol{r}(P) \\
&= \mathbf{1}.
\end{aligned}$$

Since all entries of $P$ are strictly positive, the same is true of $P_{rc}$. The claim follows.

2. The vector $\boldsymbol{r}(P)$ is a left-eigenvector of $P_{rc}$ with eigenvalue 1:

$$\begin{aligned}
\boldsymbol{r}(P)^\top P_{rc} &= \boldsymbol{r}(P)^\top\mathbf{D}(\boldsymbol{r}(P))^{-1}P\mathbf{D}(\boldsymbol{c}(P))^{-1}P^\top \\
&= \mathbf{1}^\top P\mathbf{D}(\boldsymbol{c}(P))^{-1}P^\top \\
&= \boldsymbol{c}(P)^\top\mathbf{D}(\boldsymbol{c}(P))^{-1}P^\top \\
&= \mathbf{1}^\top P^\top \\
&= \boldsymbol{r}(P)^\top.
\end{aligned}$$

3. The claim holds since

$$\begin{aligned}
\mathbf{D}(\boldsymbol{r}(P))P_{rc} &= \mathbf{D}(\boldsymbol{r}(P))\mathbf{D}(\boldsymbol{r}(P))^{-1}P\mathbf{D}(\boldsymbol{c}(P))^{-1}P^\top \\
&= P\mathbf{D}(\boldsymbol{c}(P))^{-1}P^\top \\
&= P\mathbf{D}(\boldsymbol{c}(P))^{-1}P^\top\mathbf{D}(\boldsymbol{r}(P))^{-1}\mathbf{D}(\boldsymbol{r}(P)) \\
&= P_{rc}^\top\mathbf{D}(\boldsymbol{r}(P)).
\end{aligned}$$

4. The claim follows similarly as the previous claim.

5. First, notice that Claims 3 and 4 apply analogously to all powers of $P_{rc}^l$ for $l \geq 0$. Indeed, for Claim 4 we have:

$$\begin{aligned}
P_{rc}^l\mathbf{D}(\boldsymbol{r}(P))^{-1} &= P_{rc}^{l-1}P_{rc}\mathbf{D}(\boldsymbol{r}(P))^{-1} \\
&= P_{rc}^{l-1}\mathbf{D}(\boldsymbol{r}(P))^{-1}P_{rc}^\top && \text{(by Claim 4)} \\
&= P_{rc}^{l-2}P_{rc}\mathbf{D}(\boldsymbol{r}(P))^{-1}P_{rc}^\top \\
&= P_{rc}^{l-2}\mathbf{D}(\boldsymbol{r}(P))^{-1}(P_{rc}^\top)^2 && \text{(again, by Claim 4)} \\
&= \mathbf{D}(\boldsymbol{r}(P))^{-1}(P_{rc}^\top)^l. && \text{(by repeated application of Claim 4)}
\end{aligned}$$

Then, from the Neumann series $(I - \rho P_{rc})^{-1}\mathbf{D}(\boldsymbol{r}(P))^{-1} = (\sum_{l=0}^{\infty}\rho^l P_{rc}^l)\mathbf{D}(\boldsymbol{r}(P))^{-1}$, we obtain the claim by applying the above equality to each element of the sum. ∎

**Lemma A.2** (Properties of the coefficient matrix $F_{\boldsymbol{r}}(\rho)$)**.** *Let* $\rho \in [0, 1)$, $P_{rc}$ *be a row-stochastic matrix with strictly positive entries and* $\min_i \boldsymbol{r}(P)_i \geq 0$ *for all* $i \in [n]$. *Given an* $n \times n$ *matrix* $F_{\boldsymbol{r}}(\rho) = \mathbf{D}(\boldsymbol{r}(P))(I - \rho P_{rc})$, *the following hold true:*

- $F_{\boldsymbol{r}}(\rho)$ *is a symmetric, positive-definite matrix.*

- $\lambda_{\max}(F_{\boldsymbol{r}}(\rho)) \leq (1 + \rho)\boldsymbol{r}(P)_{\max}$

- $\lambda_{\min}(F_{\boldsymbol{r}}(\rho)) \geq (1 - \rho)\boldsymbol{r}(P)_{\min}.$

*Proof.* Observe that

$$\begin{aligned}
F_{\boldsymbol{r}}(\rho) = \mathbf{D}(\boldsymbol{r}(P))(I - \rho P_{rc}) &= \mathbf{D}(\boldsymbol{r}(P)) - \rho P P_c \\
&= \mathbf{D}(\boldsymbol{r}(P)) - \rho P\mathbf{D}(\boldsymbol{c}(P))^{-1}P^\top,
\end{aligned}$$

is a sum of two symmetric matrices, which is also symmetric. The eigenvalue bounds follow from the Gerschgorin Circle Theorem. Since the smallest eigenvalue is positive, $F_{\boldsymbol{r}}(\rho)$ is positive-definite. ∎

Before we move on to the proof of Thm. 3.1, we state the following Lemma, which follows immediately from Thm. 2.7 of Fill (1991) given that $P_{rc}$ is reversible by Claim 3 of Lemma A.1.

**Lemma A.3** (Convergence to the stationary distribution under $P_{rc}$). *Given some $r \in \Delta_n$,*

$$\left\| (P_{rc}^\top)^l r - r(P) \right\|_1^2 \le \lambda_2^{2l} \chi^2(r | r(P)), \tag{18}$$

*where $\lambda_2 < 1$ is the second largest eigenvalue $\lambda_2$ of $P_{rc}$.*

Now, we provide a more formal version of Thm. 3.1 presented in the main text followed by a proof.

**Theorem 3.1** (continuing from p. 5). *Assuming $c = c(P)$ and $d_v = -P_c d_u$, define residuals $e_u(\rho) := F_r(\rho)d_u + \nabla_u g$, and $e := \nabla^2 g\, d + \nabla g$. Further, let $\lambda_2 \in (0,1)$ be the 2nd largest eigenvalue of $P_{rc}$ and $\zeta := \|\nabla_u g\|_1 / \chi(r | r(P)) \le 1$. For any $\beta \in (0,1)$, suppose:*

$$\max\left( 0, 1 - \frac{(1-\lambda_2)K}{\lambda_2(1-K)} \right) \le \rho < 1, \tag{19}$$

$$\|e_u(\rho)\|_1 \le \frac{1-\beta}{2} \eta \|\nabla g\|_1, \tag{20}$$

*where $K = \zeta\beta\eta < 1$. Then,*

$$\|e\|_1 = \|e_u(1)\|_1 \le \eta \|\nabla g\|_1. \tag{21}$$

*Proof.* To establish necessary conditions for bounding $\|e\|_1$, first, we write it out explicitly:

$$
\begin{aligned}
e &= \nabla^2 g d + \nabla g \\
&= \begin{pmatrix} \mathbf{D}(r(P)) & P \\ P^\top & \mathbf{D}(c(P)) \end{pmatrix} \begin{pmatrix} d_u \\ d_v \end{pmatrix} + \begin{pmatrix} \nabla_u g \\ \nabla_v g \end{pmatrix}, \\
&= \begin{pmatrix} \mathbf{D}(r(P)) & P \\ P^\top & \mathbf{D}(c(P)) \end{pmatrix} \begin{pmatrix} d_u \\ -P_c d_u \end{pmatrix} + \begin{pmatrix} \nabla_u g \\ 0 \end{pmatrix}, & \text{(by construction)} \\
&= \begin{pmatrix} \mathbf{D}(r(P))d_u - PP_c d_u \\ 0 \end{pmatrix} + \begin{pmatrix} \nabla_u g \\ 0 \end{pmatrix} & \text{(since } \mathbf{D}(c(P))P_c = P^\top \text{ by definition.)} \\
&= \begin{pmatrix} \mathbf{D}(r(P))(I - P_{rc})d_u \\ 0 \end{pmatrix} + \begin{pmatrix} \nabla_u g \\ 0 \end{pmatrix}, \tag{22} \\
&= \begin{pmatrix} e_u(1) \\ 0 \end{pmatrix}
\end{aligned}
$$

which proves the equality on the LHS of (21), where the last equality holds given the definitions $e_u(\rho) := F_r(\rho)d_u + \nabla_u g$ and $F_r(\rho) = \mathbf{D}(r(P))(I - \rho P_{rc})$. Hence, bounding $\|e_u(1)\|_1$ suffices. Now, given the definition of $e_u(\rho)$ and the invertibility of $F_r(\rho)$ for $\rho < 1$ by Lemma A.2, we have:

$$d_u = F_r(\rho)^{-1}(e_u(\rho) - \nabla_u g).$$

Plugging into the top half of (22), we observe that:

$$
\begin{aligned}
e_u(1) &= F_r(1)F_r(\rho)^{-1}(e_u(\rho) - \nabla_u g) + \nabla_u g \\
&= \widehat{I}(\rho)e_u(\rho) + (I - \widehat{I}(\rho))\nabla_u g,
\end{aligned}
$$

where we defined $\widehat{I}(\rho) := F_r(1)F_r(\rho)^{-1}$. Then, we have

$$\|e_u(1)\|_1 \le \left\| \widehat{I}(\rho) \right\|_1 \|e_u(\rho)\|_1 + \left\| (I - \widehat{I}(\rho))\nabla_u g \right\|_1. \tag{23}$$

First, we prove that the operator norm $\left\| \widehat{I}(\rho) \right\|_1 \le 2$.

$$
\begin{aligned}
\widehat{I}(\rho) &= \mathbf{D}(r(P))(I - P_{rc})(I - \rho P_{rc})^{-1}\mathbf{D}(r(P))^{-1} \\
&= I - (1-\rho)\mathbf{D}(r(P))P_{rc}(I - \rho P_{rc})^{-1}\mathbf{D}(r(P))^{-1} \\
&= I - (1-\rho)P_{rc}^\top(I - \rho P_{rc}^\top)^{-1}, \tag{24}
\end{aligned}
$$

where the last equality follows from claims 3 and 5 of Lemma A.1. Recalling that $\|A\|_1$ is the maximum absolute column sum of matrix $A$:

$$
\begin{aligned}
\left\|\widehat{I}(\rho)\right\|_1 &\leq 1 + (1-\rho)\left\|P_{rc}^\top\right\|_1\left\|(I - \rho P_{rc}^\top)^{-1}\right\|_1 \\
&= 1 + (1-\rho)\left\|(I - \rho P_{rc}^\top)^{-1}\right\|_1 \\
&= 1 + (1-\rho)\left\|\sum_{l=0}^\infty \rho^l (P_{rc}^\top)^l\right\|_1 \\
&\leq 1 + (1-\rho)\sum_{l=0}^\infty \rho^l \left\|(P_{rc}^\top)^l\right\|_1 \\
&= 2. \qquad\qquad \text{(since } P_{rc}^l \text{ is a stochastic matrix for all } l \geq 0)
\end{aligned}
$$

Hence, (23) simplifies to

$$
\|e_u(1)\|_1 \leq 2\|e_u(\rho)\|_1 + \left\|(I - \widehat{I}(\rho))\nabla_u g\right\|_1. \tag{25}
$$

Next, we turn to the second term on the RHS:

$$
\begin{aligned}
(I - \widehat{I}(\rho))\nabla_u g &= (1-\rho)P_{rc}^\top (I - \rho P_{rc}^\top)^{-1}\nabla_u g \qquad\qquad \text{(From (24))} \\
&= (1-\rho)\sum_{l=0}^\infty \rho^l (P_{rc}^\top)^{l+1}(r(P) - r) \\
&= (1-\rho)\sum_{l=0}^\infty \rho^l \left(r(P) - (P_{rc}^\top)^{l+1} r\right),
\end{aligned}
$$

where the last equality is due to the fact that $r(P)$ is the stationary distribution of $P_{rc}$ by Claim 2 of Lemma A.1. Then,

$$
\begin{aligned}
\left\|(I - \widehat{I}(\rho))\nabla_u g\right\|_1 &\leq (1-\rho)\left\|\sum_{l=0}^\infty \rho^l \left(r(P) - (P_{rc}^\top)^{l+1} r\right)\right\|_1 \\
&\leq (1-\rho)\sum_{l=0}^\infty \rho^l \left\|\left(r(P) - (P_{rc}^\top)^{l+1} r\right)\right\|_1 \\
&\leq (1-\rho)\sum_{l=0}^\infty \rho^l \lambda_2^{l+1}\chi(r|r(P)) \qquad\qquad \text{(By Lemma A.3)} \\
&= \frac{(1-\rho)\lambda_2}{1-\rho\lambda_2}\chi(r|r(P))
\end{aligned}
$$

Plugging this bound back into (25) and continuing with the main conditions given in the theorem:

$$
\begin{aligned}
\|e_u(1)\|_1 &\leq 2\|e_u(\rho)\|_1 + \frac{(1-\rho)\lambda_2}{1-\rho\lambda_2}\chi(r|r(P)) \\
&\leq (1-\beta)\eta\|\nabla g\|_1 + \frac{(1-\rho)\lambda_2}{1-\rho\lambda_2}\chi(r|r(P)) \qquad \text{(Since (20) holds by construction)} \\
&\leq (1-\beta)\eta\|\nabla g\|_1 + \beta\eta\|\nabla g\|_1 \qquad\qquad\quad \text{(Since (19) holds by construction)} \\
&= \eta\|\nabla g\|_1,
\end{aligned}
$$

which concludes the proof. Above, the parameter $\beta \in (0,1)$ controls a trade-off between how precisely the discounted system is solved and how aggressively the original system is discounted. In Algorithm 2 of the main text, $\beta$ was fixed at $1/2$ for simplicity. Also, for intuition on the effect of $\lambda_2$, observe that the second term vanishes as $\lambda_2 \to 0$ (if $P_{rc}$ mixes quickly) so that the Hessian can be discounted more aggressively with a smaller $\rho$. As we see next, this improves our guarantees on the condition number of the linear system. ∎

A.1.4   PROOF OF THM. 3.2

**Lemma A.4** (Spectrum after preconditioning). *Let $\widehat{F}_{\boldsymbol{r}}(\rho) := M^{-1/2} F_{\boldsymbol{r}}(\rho) M^{-1/2}$ be the diagonally preconditioned coefficient matrix, where $M = \mathbf{D}(\text{diag}(F_{\boldsymbol{r}}(\rho)))$. Further, let $\boldsymbol{\mu} = \text{diag}(P_{\boldsymbol{rc}})$ be the diagonal entries of the stochastic matrix $P_{\boldsymbol{rc}} \in \mathbb{R}_{>0}^{n \times n}$. Then, eigenvalues $\lambda_i(\rho)$ of $\widehat{F}_{\boldsymbol{r}}(\rho)$ satisfy*

$$1 - \frac{\rho(1 - \mu_{\min})}{1 - \rho\mu_{\min}} \le \lambda_i(\rho) \le 1 + \frac{\rho(1 - \mu_{\min})}{1 - \rho\mu_{\min}}, \ \forall i \in [n] \tag{26}$$

$$\kappa(\rho) = \frac{\lambda_{\max}(\rho)}{\lambda_{\min}(\rho)} \le 2 \left( \mu_{\min} + \frac{1 - \mu_{\min}}{1 - \rho} \right) \le \frac{2}{1 - \rho} \tag{27}$$

*Proof.* The proof of (26) follows straightforwardly from the Gerschgorin Circle Theorem. Due to diagonal similarity, the spectrum of $\widehat{F}_{\boldsymbol{r}}(\rho)$ coincides with that of $\widetilde{F}_{\boldsymbol{r}}(\rho) = M^{-1} F_{\boldsymbol{r}}(\rho)$. Clearly, diagonal entries of $\widetilde{F}_{\boldsymbol{r}}(\rho)$ all equal to 1, so that all Gerschgorin disks are centered around unity. Then, all eigenvalues must be inside the biggest disk, which contains all of the smaller disks. First, consider row $i$ of $F_{\boldsymbol{r}}(\rho) = \mathbf{D}(\boldsymbol{r}(P))(I - \rho P_{\boldsymbol{rc}})$:

$$\sum_{j \neq i} |\widetilde{F}_{\boldsymbol{r}}(\rho)_{ij}| = \boldsymbol{r}(P)_i \rho(1 - \mu_i).$$

Since $M_{ii} = F_{\boldsymbol{r}}(\rho)_{ii} = \boldsymbol{r}(P)_i(1 - \rho\mu_i)$, we then have:

$$\sum_{j \neq i} |\widetilde{F}_{\boldsymbol{r}}(\rho)_{ij}| = \frac{\rho(1 - \mu_i)}{(1 - \rho\mu_i)},$$

where the biggest Gerschgorin disk corresponds to $\mu_{\min}$, so that (26) holds for all $i \in [n]$. Then,

$$\begin{aligned}
\kappa(\rho) &\le \frac{2}{\lambda_{\min}(\rho)} \le \frac{2}{1 - \frac{\rho(1 - \mu_{\min})}{(1 - \rho\mu_{\min})}} \\
&= 2 \left( \mu_{\min} + \frac{1 - \mu_{\min}}{1 - \rho} \right) \\
&= O \left( \frac{1 - \mu_{\min}}{1 - \rho} \right) \quad \text{as } \rho \to 1. \quad \blacksquare
\end{aligned}$$

**Lemma A.5** (Equivalence of norms). *Suppose $\boldsymbol{d}_{\boldsymbol{u}}^*$ satisfies $F_{\boldsymbol{r}}(\rho)\boldsymbol{d}_{\boldsymbol{u}}^* = -\nabla_{\boldsymbol{u}} g$. Let $\widehat{F}_{\boldsymbol{r}}(\rho) = M^{-1/2} F_{\boldsymbol{r}}(\rho) M^{-1/2}$ and $M = \mathbf{D}(\text{diag}(F_{\boldsymbol{r}}(\rho)))$ as in Lemma A.4. Define the reparametrization $\widehat{\boldsymbol{d}}_{\boldsymbol{u}} = M^{1/2} \boldsymbol{d}_{\boldsymbol{u}}$ given some $\boldsymbol{d}_{\boldsymbol{u}} \in \mathbb{R}^n$, and the residual $\boldsymbol{e}_{\boldsymbol{u}} = F_{\boldsymbol{r}}(\rho)\boldsymbol{d}_{\boldsymbol{u}} + \nabla_{\boldsymbol{u}} g$. We have,*

$$\sqrt{(1 - \rho)\boldsymbol{r}(P)_{\min}} \le \frac{\|\boldsymbol{e}_{\boldsymbol{u}}\|_1}{\left\| \widehat{\boldsymbol{d}}_{\boldsymbol{u}} - \widehat{\boldsymbol{d}}_{\boldsymbol{u}}^* \right\|_{\widehat{F}_{\boldsymbol{r}}(\rho)}} \le \sqrt{(1 + \rho)n}. \tag{28}$$

*Proof.* For this proof, we drop $\rho$ from $F_{\boldsymbol{r}}(\rho)$ and $\widehat{F}_{\boldsymbol{r}}(\rho)$ for convenience. First, observe that $\boldsymbol{d_u} - \boldsymbol{d_u^*} = F_{\boldsymbol{r}}^{-1}\boldsymbol{e_u}$. Then,

$$
\begin{aligned}
\left\| \widehat{\boldsymbol{d}}_{\boldsymbol{u}} - \widehat{\boldsymbol{d}}_{\boldsymbol{u}}^* \right\|_{\widehat{F}_{\boldsymbol{r}}}^2 &= \left\| M^{1/2} F_{\boldsymbol{r}}^{-1} \boldsymbol{e_u} \right\|_{\widehat{F}_{\boldsymbol{r}}}^2 \\
&= \boldsymbol{e_u}^\top F_{\boldsymbol{r}}^{-1} M^{1/2} \widehat{F}_{\boldsymbol{r}} M^{1/2} F_{\boldsymbol{r}}^{-1} \boldsymbol{e_u} \\
&= \boldsymbol{e_u}^\top F_{\boldsymbol{r}}^{-1} F_{\boldsymbol{r}} F_{\boldsymbol{r}}^{-1} \boldsymbol{e_u} \\
&= \boldsymbol{e_u}^\top F_{\boldsymbol{r}}^{-1} \boldsymbol{e_u} \\
&\geq \lambda_{\min}(F_{\boldsymbol{r}}^{-1}) \left\| \boldsymbol{e_u} \right\|_2^2 \\
&= \frac{\left\| \boldsymbol{e_u} \right\|_2^2}{\lambda_{\max}(F_{\boldsymbol{r}})} \\
&\geq \frac{\left\| \boldsymbol{e_u} \right\|_2^2}{(1+\rho)\boldsymbol{r}(P)_{\max}} && \text{(by Lemma A.2)} \\
&\geq \frac{\left\| \boldsymbol{e_u} \right\|_2^2}{(1+\rho)} \\
&\geq \frac{\left\| \boldsymbol{e_u} \right\|_1^2}{n(1+\rho)}, && (\text{since } \left\| \boldsymbol{x} \right\|_1 \leq \sqrt{n} \left\| \boldsymbol{x} \right\|_2, \forall \boldsymbol{x} \in \mathbb{R}^n.)
\end{aligned}
$$

which is equivalent to the upper bound of the desired result. The lower bound follows similarly in the reverse direction, using $\lambda_{\min}(F_{\boldsymbol{r}})$ and $\left\| \boldsymbol{x} \right\|_2 \leq \left\| \boldsymbol{x} \right\|_1$. ∎

**Lemma A.6** (Convergence of CG). *Suppose diagonally-preconditioned conjugate gradient method is initialized with $\boldsymbol{d}_{\boldsymbol{u}}^{(0)} = \boldsymbol{0}$ for the linear system $F_{\boldsymbol{r}}(\rho)\boldsymbol{d}_{\boldsymbol{u}}^* = -\nabla_{\boldsymbol{u}}g$, where $\rho \in [0,1)$. Let $\mu$ be the largest diagonal entry of $P_{\boldsymbol{rc}}$. Assuming that $\boldsymbol{r}(P)_{\min} \geq \varepsilon_{\mathrm{d}}/(4n)$ for all $i \in [n]$ given some constant $\varepsilon_{\mathrm{d}} > 0$, the residual satisfies $\|F_{\boldsymbol{r}}(\rho)\boldsymbol{d_u} + \nabla_{\boldsymbol{u}}g\|_1 = \|\boldsymbol{e_u}(\rho)\|_1 \leq \hat{\eta} \|\nabla_{\boldsymbol{u}}g\|_1$ after at most* $\mathrm{ceil}(k)$ *steps, where*

$$
k \leq \left( \frac{1-\mu\rho}{1-\rho} \right)^{1/2} \log\left( 6n(1-\rho)^{-1/2}\hat{\eta}^{-1}\varepsilon_{\mathrm{d}}^{-1/2} \right) = \widetilde{O}\left( \sqrt{\frac{1-\mu}{1-\rho}} \right) \text{ as } \rho \to 1. \tag{29}
$$

*Proof.* Once again, we drop $\rho$ from $F_{\boldsymbol{r}}(\rho)$ and $\widehat{F}_{\boldsymbol{r}}(\rho)$ for convenience. Using the same definitions as Lemma A.5, recall the equivalence of the diagonally-preconditioned linear system:

$$
\begin{aligned}
F_{\boldsymbol{r}}\boldsymbol{d_u} &= -\nabla_{\boldsymbol{u}}g \\
\Longleftrightarrow \quad M^{-1/2}F_{\boldsymbol{r}}\boldsymbol{d_u} &= -M^{-1/2}\nabla_{\boldsymbol{u}}g \\
\Longleftrightarrow \quad M^{-1/2}F_{\boldsymbol{r}}M^{-1/2}M^{1/2}\boldsymbol{d_u} &= -M^{-1/2}\nabla_{\boldsymbol{u}}g \\
\Longleftrightarrow \quad \widehat{F}_{\boldsymbol{r}}\widehat{\boldsymbol{d}}_{\boldsymbol{u}} &= -M^{-1/2}\nabla_{\boldsymbol{u}}g.
\end{aligned}
$$

To guarantee $\|\boldsymbol{e_u}\|_1 \leq \hat{\eta} \|\nabla_{\boldsymbol{u}}g\|_1$, it is sufficient to have (given the upper bound in Lemma A.5)

$$
\left\| \widehat{\boldsymbol{d}}_{\boldsymbol{u}} - \widehat{\boldsymbol{d}}_{\boldsymbol{u}}^* \right\|_{\widehat{F}_{\boldsymbol{r}}(\rho)} \leq \frac{\hat{\eta} \|\nabla_{\boldsymbol{u}}g\|_1}{\sqrt{n(1+\rho)}} := 2\varepsilon. \tag{30}
$$

It is well-known the that linear conjugate gradient method ensures $\left\| \widehat{\boldsymbol{d}}_{\boldsymbol{u}} - \widehat{\boldsymbol{d}}_{\boldsymbol{u}}^* \right\|_{\widehat{F}_{\boldsymbol{r}}(\rho)} \leq 2\varepsilon$ after at most $\mathrm{ceil}(k)$ steps (Shewchuk, 1994), where

$$
k = \frac{1}{2}\sqrt{\kappa(\widehat{F}_{\boldsymbol{r}})} \log \left\| \widehat{\boldsymbol{d}}_{\boldsymbol{u}}^{(0)} - \widehat{\boldsymbol{d}}_{\boldsymbol{u}}^* \right\|_{\widehat{F}_{\boldsymbol{r}}} \varepsilon^{-1}. \tag{31}
$$

Then, we have

$$
\begin{aligned}
k &\le \frac{1}{2}\sqrt{\kappa(\widehat{F}_{\bm{r}})}\log\left\|\widehat{\bm{d}}_{\bm{u}}^{(0)}-\widehat{\bm{d}}_{\bm{u}}^{*}\right\|_{\hat{F}_{\bm{r}}}\varepsilon^{-1} \\
&\le \frac{1}{2}\sqrt{\kappa(\widehat{F}_{\bm{r}})}\log\frac{\left\|\bm{e}_{\bm{u}}^{(0)}\right\|_{1}\varepsilon^{-1}}{\sqrt{(1-\rho)\bm{r}(P)_{\min}}} && \text{(by the lower bound in Lemma A.5)}\\
&= \frac{1}{2}\sqrt{\kappa(\widehat{F}_{\bm{r}})}\log\frac{\|\nabla_{\bm{u}}g\|_{1}\varepsilon^{-1}}{\sqrt{(1-\rho)\bm{r}(P)_{\min}}} && \left(\text{since }\bm{d}_{\bm{u}}^{(0)}=\bm{0}\implies\left\|\bm{e}_{\bm{u}}^{(0)}\right\|_{1}=\|\nabla_{\bm{u}}g\|_{1}.\right)\\
&= \frac{1}{2}\sqrt{\kappa(\widehat{F}_{\bm{r}})}\log 2\hat{\eta}^{-1}\sqrt{\frac{(1+\rho)n}{(1-\rho)\bm{r}(P)_{\min}}} && \text{(using }\varepsilon\text{ from (30))}\\
&\le \frac{1}{2}\sqrt{\kappa(\widehat{F}_{\bm{r}})}\log 3n^{1/2}(1-\rho)^{-1/2}\hat{\eta}^{-1}\bm{r}(P)_{\min}^{-1/2} && \text{(simplifying constants)}\\
&\le \frac{1}{2}\sqrt{\kappa(\widehat{F}_{\bm{r}})}\log 6n(1-\rho)^{-1/2}\hat{\eta}^{-1}\varepsilon_{\mathrm{d}}^{-1/2}, && \text{(since }\bm{r}(P)_{\min}\ge\varepsilon_{\mathrm{d}}/(4n)\text{ by assumption)}\\
&\le \sqrt{\frac{1-\mu\rho}{2(1-\rho)}}\log 6n(1-\rho)^{-1/2}\hat{\eta}^{-1}\varepsilon_{\mathrm{d}}^{-1/2} && \text{(by Lemma A.4)}
\end{aligned}
$$

which concludes the proof. $\blacksquare$

**Theorem 3.2** (Convergence of Algorithm 2). *Suppose $\bm{r}(P)_{\min}\ge\varepsilon_{\mathrm{d}}/(4n)$ given $\varepsilon_{\mathrm{d}}>0$ and each step of Alg. 2 runs diagonally-preconditioned CG initialized with $\bm{d}_{\bm{u}}^{(0)}{=}\bm{0}$. Alg. 2 terminates in*

$$
\widetilde{O}\left(n^{2}\sqrt{\frac{(1-\mu)\chi(\bm{r}|\bm{r}(P))}{(1-\lambda_{2})\eta\|\bm{r}(P)-\bm{r}\|_{1}}}\log\varepsilon_{\mathrm{d}}^{-1/2}\right) \tag{14}
$$

*operations, where $\lambda_{2}<1$ is the $2^{nd}$ largest eigenvalue of $P_{\bm{rc}}$ and $\mu<1$ its smallest diagonal entry.*

*Proof.* An easy way to see why Thm. 3.2 holds is by noticing that the lower bound condition on $\rho$ given in (19) will hold after a finite number of iterations depending logarithmically on remaining problem parameters. Then, since by Lemma A.6 each iteration requires $\widetilde{O}((1-\rho)^{-1/2})$ steps of CG (with each step costing $O(n^{2})$), and $1-\rho$ will be at most $O(1-\lambda_{2})$, the result follows. Now, we provide a more detailed analysis.

First, define $\tilde{\rho}^{(l)}=1-\rho^{(l)}$ for the $l^{th}$ iteration of Alg. 2. Since $\rho^{(0)}=0\implies\tilde{\rho}^{(0)}=1$, we have $\tilde{\rho}^{(l)}=4^{-l}$ given the update rule in L5. Then, by (19) of Thm. 3.1, Alg. 1 performs the final iteration (in the worst case) when

$$
\tilde{\rho}^{(l)}=4^{-l}\le\frac{(1-\lambda_{2})K}{\lambda_{2}(1-K)}=(1-\lambda_{2})\tilde{K}.
$$

That is, in the worst case we terminate after $l$ steps, where

$$
l=\mathrm{ceil}\left(-\frac{\log(1-\lambda_{2})\tilde{K}}{\log(4)}\right). \tag{32}
$$

In the worst case, we are guaranteed by the final step final that,

$$
\tilde{\rho}^{(l)}=4^{-l}\in\left(\frac{(1-\lambda_{2})\tilde{K}}{4},(1-\lambda_{2})\tilde{K}\right]. \tag{33}
$$

Thus, for each integer $l'\in[l]$, we have

$$
\tilde{\rho}^{(l')}=4^{-l'}>(1-\lambda_{2})\tilde{K}4^{l-l'-1}. \tag{34}
$$

Then, the condition number at a given step $l' \geq 0$ satisfies:

$$\kappa^{(l')} \leq \mu + \frac{1-\mu}{1-\rho^{(l')}} \qquad \text{(From Lemma A.4)}$$

$$\leq \mu + 4^{1+l'-l}\frac{1-\mu}{(1-\lambda_2)\tilde{K}} \qquad \text{(From (34))}$$

$$= 4^{1+l'-l}\kappa_0,$$

where

$$\kappa_0 = O\left(\frac{1-\mu}{(1-\lambda_2)\tilde{K}}\right) \text{ as } \lambda_2 \to 1 \qquad (35)$$

By Lemma A.6, a given step $l' \leq l$ of Alg. 2 takes at most

$$k(l') = 2^{1+l'-l}\sqrt{\kappa_0}\log\left(3n\sqrt{\kappa_0}2^{1+l'-l}\hat{\eta}^{-1}\varepsilon_{\mathrm{d}}^{-1/2}\right)$$

$$\leq 2^{1+l'-l}\sqrt{\kappa_0}\log\left(6n\sqrt{\kappa_0}\hat{\eta}^{-1}\varepsilon_{\mathrm{d}}^{-1/2}\right) \qquad \text{(since } l' \leq l\text{)}$$

$$= 2^{1+l'-l}\widetilde{O}(\sqrt{\kappa_0}),$$

steps of the conjugate gradient algorithm. Taking a sum over $l'$:

$$\sum_{l'=0}^{l} k(l') = \widetilde{O}(\sqrt{\kappa_0})2^{1-l}\sum_{l'=0}^{l}2^{l'}$$

$$= \widetilde{O}(\sqrt{\kappa_0})2^{1-l}\left(\frac{2^{l+1}-1}{2-1}\right) \qquad (36)$$

$$= \widetilde{O}(\sqrt{\kappa_0})(4 - 2^{1-l})$$

$$\leq \widetilde{O}(\sqrt{\kappa_0})(4 - \sqrt{(1-\lambda_2)\tilde{K}}) \qquad \text{(due to (32))}$$

$$= \widetilde{O}(\sqrt{\kappa_0}). \qquad (37)$$

Note that the choice of decay factor 4 is not arbitrary; if one chooses instead a factor $L > 1$ and repeats the above steps, one arrives at (cf. (32)):

$$l = \mathrm{ceil}\left(-\frac{\log(1-\lambda_2)\tilde{K}}{\log(L)}\right),$$

and the following (cf. (36))

$$\sum_{l'=0}^{l} k(l') = \widetilde{O}(\sqrt{\kappa_0})\sqrt{L}^{1-l}\left(\frac{\sqrt{L}^{l+1}-1}{\sqrt{L}-1}\right)$$

$$\leq \widetilde{O}(\sqrt{\kappa_0})\frac{L - \sqrt{(1-\lambda_2)\tilde{K}}}{\sqrt{L}-1}$$

$$\leq \widetilde{O}(\sqrt{\kappa_0})\frac{L}{\sqrt{L}-1},$$

and $L = 4$ is the global minimizer of $h(x) = x/(\sqrt{x}-1)$. The last inequality is strong when convergence is bottlenecked by $\lambda_2 \approx 1$, so that the choice $L = 4$ becomes near-optimal when optimality is most needed.

Continuing from (37), the overall complexity is:

$$\tilde{O}(\sqrt{\kappa_0}) = \tilde{O}\left(\sqrt{\frac{1-\mu}{(1-\lambda_2)\tilde{K}}}\right).$$

The result follows by explicitly writing and simplifying $\tilde{K}^{-1}$:

$$
\begin{aligned}
\tilde{K}^{-1} &= \frac{\lambda_2(1-K)}{K} \\
&= \frac{2\lambda_2(1-\zeta\eta/2)}{\zeta\eta}. \qquad \text{(Since we choose } \beta = 0.5, \text{ where } K = \zeta\beta\eta) \\
&\leq \frac{2}{\zeta\eta} \\
&= 2\frac{\chi(\boldsymbol{r}|\boldsymbol{r}(P))}{\|\nabla_{\boldsymbol{u}}g\|_1 \eta},
\end{aligned}
$$

where the last equality holds by definition of $\zeta$. ∎

### A.1.5 Proofs of Lemma 3.3 and Corollary 3.4

**Lemma 3.3** (Convergence of Algorithm 3). *Assuming that $\|\boldsymbol{r}(P)/\boldsymbol{r}\|_\infty < \infty$ and $\|\boldsymbol{r}/\boldsymbol{r}(P)\|_\infty < \infty$, Algorithm 3 converges in $O(n^2/\varepsilon_\chi)$ operations.*

*Proof.* First, recall that by Eq. (169) of Sason & Verdú (2016), the ratio $\frac{D_{\mathrm{KL}}(\boldsymbol{r}|\boldsymbol{r}(P))}{\chi^2(\boldsymbol{r}|\boldsymbol{r}(P))}$ is bounded both above and below by constants depending on $\|\boldsymbol{r}(P)/\boldsymbol{r}\|_\infty < \infty$ and $\|\boldsymbol{r}/\boldsymbol{r}(P)\|_\infty < \infty$. Notably, the ratio converges to $1/2$ as $\boldsymbol{r}(P) \to \boldsymbol{r}$ as shown in Thm. 4.1 of (Csiszár & Shields, 2004).

Since we know that after $k$ steps of Sinkhorn iteration,

$$D_{\mathrm{KL}}(\boldsymbol{r}|\boldsymbol{r}(P)) \leq A/k$$

for some constant $A > 0$ (see Corollary 6.12 of Nutz (2021)) and $\chi^2(\boldsymbol{r}|\boldsymbol{r}(P)) = O(D_{\mathrm{KL}}(\boldsymbol{r}|\boldsymbol{r}(P)))$, we conclude that after $k$ steps, $\chi^2(\boldsymbol{r}|\boldsymbol{r}(P)) = O(k^{-1})$. The result follows as each Sinkhorn iteration costs $O(n^2)$ and Alg. 2 terminates when $\chi^2(\boldsymbol{r}|\boldsymbol{r}(P)) \leq \varepsilon_\chi$. ∎

**Corollary 3.4** (Per-step Cost of Algorithm 4). *If the backtracking line search in Alg. 4 converges in $S$ iterations, then an iteration of Alg. 4 costs $\widetilde{O}(n^2(S + \varepsilon_{\mathrm{d}}^{-2/5}(1-\lambda_2)^{-1/2}))$ operations, where $\lambda_2 < 1$ is the 2nd largest eigenvalue of $P_{\boldsymbol{rc}}$ defined as in (9) and evaluated at $\boldsymbol{u}, \boldsymbol{v}$ (cf. (3)).*

*Proof.* First, from Lemma 3.3, the ChiSinkhorn routine in L6 of Alg. 4 has complexity $O(n^2\varepsilon_{\mathrm{d}}^{-2/5})$ as we chose $\varepsilon_\chi = \varepsilon_{\mathrm{d}}^{2/5}$. Lines L9 and L18 each cost $O(n^2)$, which leaves the cost of line search between lines L12-14 and the NewtonSolve routine in L8. Since the former is assumed to take $S$ steps, each costing $O(n^2)$, it remains to show the cost of NewtonSolve (or Alg. 2).

First, consider the case when $\eta = 0.4\varepsilon_{\mathrm{d}}/\|\nabla_{\boldsymbol{u}}g\|_1$ as per L7. From Thm. 3.2, after dropping the logarithmic terms and the linear $n^2$ term, we have the following linear term:

$$
\sqrt{\frac{(1-\mu)\chi(\boldsymbol{r}|\boldsymbol{r}(P))}{(1-\lambda_2)\eta\|\boldsymbol{r}(P)-\boldsymbol{r}\|_1}} \leq \sqrt{\frac{(1-\mu)\chi(\boldsymbol{r}|\boldsymbol{r}(P))}{(1-\lambda_2)\varepsilon_{\mathrm{d}}}} \leq \sqrt{\frac{(1-\mu)\varepsilon_{\mathrm{d}}^{-4/5}}{(1-\lambda_2)}} = O(\varepsilon_{\mathrm{d}}^{-2/5}(1-\lambda_2)^{-1/2})
$$

since $\chi^2(\boldsymbol{r}|\boldsymbol{r}(P)) \leq \varepsilon_{\mathrm{d}}^{2/5}$. Next, consider the case when $\|\nabla_{\boldsymbol{u}}g\|_1 > 0.4\varepsilon_{\mathrm{d}}/\|\nabla_{\boldsymbol{u}}g\|_1$, i.e., $\|\nabla_{\boldsymbol{u}}g\|_1^2 > 0.4\varepsilon_{\mathrm{d}}$, so that L7 assigns $\eta = \|\nabla_{\boldsymbol{u}}g\|_1$ instead. Inserting $\eta$ into the denominator on the LHS above and applying $\|\nabla_{\boldsymbol{u}}g\|_1^2 > 0.4\varepsilon_{\mathrm{d}}$ yields the same complexity. ∎

### A.1.6 Proof of Thm. 3.5

First, we present a simple lemma on the line search condition used in L12 of Algorithm 4.

**Lemma A.7** (Sufficient Decrease Condition). *Assuming $\boldsymbol{c} = \boldsymbol{c}(P)$, given a descent direction $\boldsymbol{d} = (\boldsymbol{d_u}, -P_{\boldsymbol{c}}\boldsymbol{d_u})$, the Armijo condition for line search over objective $g$ is satisfied for a step size $\alpha \in (0, 1]$ and constant parameter $c_1 \in (0, 1)$ if and only if:*

$$\|\mathbf{vec}(P(\boldsymbol{u}+\alpha\boldsymbol{d_u}, \boldsymbol{v}-\alpha P_{\boldsymbol{c}}\boldsymbol{d_u}))\|_1 - 1 \leq (1-c_1)\alpha\langle-\nabla_{\boldsymbol{u}}g, \boldsymbol{d_u}\rangle. \tag{38}$$

*Proof.* Recall that the Armijo condition requires (see Ch. 3.1 of Nocedal & Wright (2006)):

$$g(\boldsymbol{u} + \alpha\boldsymbol{d_u}, \boldsymbol{v} - \alpha P_{\boldsymbol{c}}\boldsymbol{d_u}) \leq g(\boldsymbol{u}, \boldsymbol{v}) + c_1\alpha\langle\nabla g, \boldsymbol{d}\rangle. \tag{39}$$

For brevity, let $P(\alpha) := P(\boldsymbol{u} + \alpha\boldsymbol{d_u}, \boldsymbol{v} - \alpha P_{\boldsymbol{c}}\boldsymbol{d_u})$. We show the equivalence of the above statement to (38) step by step:

$$g(\boldsymbol{u} + \alpha\boldsymbol{d_u}, \boldsymbol{v} - \alpha P_{\boldsymbol{c}}\boldsymbol{d_u}) \leq g(\boldsymbol{u}, \boldsymbol{v}) + c_1\alpha\langle\nabla g, \boldsymbol{d}\rangle.$$

$$\iff \|\mathbf{vec}(P(\alpha))\|_1 - 1 - \langle\boldsymbol{u} + \alpha\boldsymbol{d_u}, \boldsymbol{r}\rangle - \langle\boldsymbol{v} - \alpha P_{\boldsymbol{c}}\boldsymbol{d_u}, \boldsymbol{c}\rangle \leq -\langle\boldsymbol{u}, \boldsymbol{r}\rangle - \langle\boldsymbol{v}, \boldsymbol{c}\rangle + c_1\alpha\langle\nabla g, \boldsymbol{d}\rangle$$

$$\iff \|\mathbf{vec}(P(\alpha))\|_1 - 1 \leq \alpha\big(\langle\boldsymbol{d_u}, \boldsymbol{r}\rangle - \langle P_{\boldsymbol{c}}\boldsymbol{d_u}, \boldsymbol{c}\rangle\big) + c_1\alpha\langle\nabla g, \boldsymbol{d}\rangle$$

$$\iff \|\mathbf{vec}(P(\alpha))\|_1 - 1 \leq \alpha\big(\langle\boldsymbol{d_u}, \boldsymbol{r}\rangle - \langle\boldsymbol{d_u}, P_{\boldsymbol{c}}^\top\boldsymbol{c}\rangle\big) + c_1\alpha\langle\nabla g, \boldsymbol{d}\rangle$$

$$\iff \|\mathbf{vec}(P(\alpha))\|_1 - 1 \leq \alpha\big(\langle\boldsymbol{d_u}, \boldsymbol{r}\rangle - \langle\boldsymbol{d_u}, \boldsymbol{r}(P)\rangle\big) + c_1\alpha\langle\nabla g, \boldsymbol{d}\rangle \qquad \text{(Since } P_{\boldsymbol{c}}^\top\boldsymbol{c} = \boldsymbol{r}(P)\text{)}$$

$$\iff \|\mathbf{vec}(P(\alpha))\|_1 - 1 \leq \alpha\langle\boldsymbol{d_u}, -\nabla_{\boldsymbol{u}}g\rangle + c_1\alpha\langle\nabla_{\boldsymbol{u}}g, \boldsymbol{d_u}\rangle, \quad \text{(Since } \boldsymbol{c}(P) = \boldsymbol{c} \text{ by assumption)}$$

which is equivalent to (38). ∎

**Theorem 3.5** (Per-step Improvement of Algorithm 4). *Given a descent direction* $\boldsymbol{d} = (\boldsymbol{d_u}, -P_{\boldsymbol{c}}\boldsymbol{d_u})$ *such that* $\|\boldsymbol{e}\|_1 = \|\nabla^2 g_k \boldsymbol{d} + \nabla g_k\|_1 \leq \eta\|\nabla g_k\|_1$, *let* $\alpha \in (0, 1]$ *be the step size found via backtracking line search in the* $k^{\text{th}}$ *step of Alg. 4. Then,* $\nabla g_{k+1} := \nabla g(\boldsymbol{u} + \alpha\boldsymbol{d_u}, \boldsymbol{v} - \alpha P_{\boldsymbol{c}}\boldsymbol{d_u})$ *satisfies*

$$\|\nabla g_{k+1}\|_1 \leq (1 - \alpha + \alpha\eta)\|\nabla g_k\|_1 + \alpha\sqrt{\alpha}\, O(\|\nabla g_k\|_1^2). \tag{15}$$

*Proof.* For convenience, let $\boldsymbol{z} := (\boldsymbol{u}, \boldsymbol{v})$ given some $(\boldsymbol{u}, \boldsymbol{v}) \in \mathbb{R}^{2n}$ and $\nabla^2 g(t) := \nabla^2 g(\boldsymbol{z} + t\alpha\boldsymbol{d})$. Recall that from Taylor's Theorem, we have:

$$\nabla g(\boldsymbol{z} + \alpha\boldsymbol{d}) = \nabla g(\boldsymbol{z}) + \alpha\int_0^1 [\nabla^2 g(t)]\boldsymbol{d}\, dt$$

$$= \nabla g(\boldsymbol{z}) + \alpha\nabla^2 g(0)\boldsymbol{d} + \alpha\int_0^1 \big[\nabla^2 g(t) - \nabla^2 g(0)\big]\boldsymbol{d}\, dt$$

$$= (1 - \alpha)\nabla g(\boldsymbol{z}) + \alpha\boldsymbol{e} + \alpha\int_0^1 \big[\nabla^2 g(t) - \nabla^2 g(0)\big]\boldsymbol{d}\, dt.$$

Now, we define the following:

$$\boldsymbol{h}(t) := \int_0^1 \big[\nabla^2 g(t) - \nabla^2 g(0)\big]\boldsymbol{d}.$$

Then,

$$\|\nabla g(\boldsymbol{z} + \alpha\boldsymbol{d})\|_1 \leq (1 - \alpha)\|\nabla g(\boldsymbol{z})\|_1 + \alpha\|\boldsymbol{e}\|_1 + \alpha\left\|\int_0^1 \boldsymbol{h}(t)dt\right\|_1$$

$$\leq (1 - \alpha + \alpha\eta)\|\nabla g(\boldsymbol{z})\|_1 + \alpha\left\|\int_0^1 \boldsymbol{h}(t)dt\right\|_1 \qquad \text{(By construction)}$$

$$\leq (1 - \alpha + \alpha\eta)\|\nabla g(\boldsymbol{z})\|_1 + \alpha\int_0^1 \|\boldsymbol{h}(t)\|_1\, dt \tag{40}$$

We will bound $\|\boldsymbol{h}(t)\|_1$ in terms of $t \in [0, 1]$ and evaluate the integral. Again, define $P(t) := P(\boldsymbol{z} + t\alpha\boldsymbol{d})$ for convenience, and let $\Delta P(t) := P(t) - P$, where $P = P(0)$. Then,

$$\boldsymbol{h}(t) = \begin{pmatrix} \mathbf{D}(\boldsymbol{r}(\Delta P(t))) & \Delta P(t) \\ \Delta P(t)^\top & \mathbf{D}(\boldsymbol{c}(\Delta P(t))) \end{pmatrix} \begin{pmatrix} \boldsymbol{d_u} \\ -P_{\boldsymbol{c}}\boldsymbol{d_u} \end{pmatrix}$$

$$= \begin{pmatrix} \mathbf{D}(\boldsymbol{r}(\Delta P(t)))\boldsymbol{d_u} \\ -\mathbf{D}(\boldsymbol{c}(\Delta P(t)))P_{\boldsymbol{c}}\boldsymbol{d_u} \end{pmatrix} + \begin{pmatrix} -\Delta P(t)P_{\boldsymbol{c}}\boldsymbol{d_u} \\ \Delta P(t)^\top\boldsymbol{d_u} \end{pmatrix}$$

$$:= \begin{pmatrix} \boldsymbol{h}_1(t) \\ \boldsymbol{h}_2(t) \end{pmatrix} + \begin{pmatrix} \boldsymbol{h}_3(t) \\ \boldsymbol{h}_4(t) \end{pmatrix},$$

where we have $\|\boldsymbol{h}(t)\|_1 \le \sum_{l=1}^4 \|\boldsymbol{h}_l(t)\|_1$. Consider $\boldsymbol{h}_1(t)$,

$$\|\boldsymbol{h}_1(t)\|_1 = \sum_i |\boldsymbol{r}(\Delta P(t))_i (\boldsymbol{d_u})_i|$$
$$\le \|\boldsymbol{d_u}\|_\infty \sum_i |\boldsymbol{r}(\Delta P(t))_i|$$
$$= \|\boldsymbol{d_u}\|_\infty \|\boldsymbol{r}(\Delta P(t))\|_1$$
$$\le \|\boldsymbol{d_u}\|_\infty \|\mathbf{vec}(\Delta P(t))\|_1 .$$

Since we have $\|P_{\boldsymbol{c}} \boldsymbol{d_u}\|_\infty \le \|\boldsymbol{d_u}\|_\infty$, the same bound holds for $\|\boldsymbol{h}_2(t)\|_1$ by symmetry. Next,

$$\|\boldsymbol{h}_4(t)\|_1 = \left\|\Delta P(t)^\top \boldsymbol{d_u}\right\|_1$$
$$= \sum_j |\langle \Delta P(t)_{:j}, \boldsymbol{d_u} \rangle| \qquad (A_{:j} \text{ denotes the } j^{\text{th}} \text{ column of A.})$$
$$\le \|\boldsymbol{d_u}\|_\infty \sum_j \|\Delta P(t)_{:j}\|_1$$
$$= \|\boldsymbol{d_u}\|_\infty \|\mathbf{vec}(\Delta P(t))\|_1 .$$

Similarly, the same bound holds for $\|\boldsymbol{h}_3(t)\|_1$ by symmetry. Then,

$$\|\boldsymbol{h}(t)\|_1 \le 4 \|\boldsymbol{d_u}\|_\infty \|\mathbf{vec}(\Delta P(t))\|_1 . \tag{41}$$

From Pinsker's inequality, we have:

$$\frac{1}{2} \|\mathbf{vec}(\Delta P(t))\|_1^2 \le D_h(P|P(t)) \qquad \text{(Bregman divergence under negative entropy)}$$
$$= \|\mathbf{vec}(P(t))\|_1 - 1 + \langle P, \log(P/P(t)) \rangle \quad \text{(Given } \boldsymbol{c} = \boldsymbol{c}(P), \mathbf{vec}(P) \in \Delta_{n^2}.)$$
$$= \|\mathbf{vec}(P(t))\|_1 - 1 - t\alpha \left(\langle \boldsymbol{r}(P), \boldsymbol{d_u} \rangle + \langle \boldsymbol{c}, -P_{\boldsymbol{c}} \boldsymbol{d_u} \rangle\right)$$
$$= \|\mathbf{vec}(P(t))\|_1 - 1 - t\alpha \left(\langle \boldsymbol{r}(P), \boldsymbol{d_u} \rangle - \langle P_{\boldsymbol{c}}^\top \boldsymbol{c}, \boldsymbol{d_u} \rangle\right)$$
$$= \|\mathbf{vec}(P(t))\|_1 - 1 \qquad \text{(Since } P_{\boldsymbol{c}}^\top \boldsymbol{c} = P\mathbf{D}(\boldsymbol{c})^{-1}\boldsymbol{c} = P\mathbf{1} = \boldsymbol{r}(P).)$$
$$\le 0.99 t\alpha \langle -\nabla_{\boldsymbol{u}} g, \boldsymbol{d_u} \rangle,$$

where the last inequality is due to (38) as we assumed $\alpha$ satisfies the Armijo condition, and if step size $\alpha$ is feasible, so is any step size $t\alpha \in [0, \alpha]$ given that the objective is convex. Plugging the above into (41):

$$\|\boldsymbol{h}(t)\|_1 \le 4 \|\boldsymbol{d_u}\|_\infty \sqrt{2 * 0.99 t\alpha \langle -\nabla_{\boldsymbol{u}} g, \boldsymbol{d_u} \rangle))}$$
$$\le 4\sqrt{2\alpha} \|\boldsymbol{d_u}\|_\infty^{3/2} \|\nabla_{\boldsymbol{u}} g\|_1^{1/2} \sqrt{t}.$$

Hence,

$$\int_0^1 \|\boldsymbol{h}(t)\|_1 \, dt \le 4\sqrt{2\alpha} \|\boldsymbol{d_u}\|_\infty^{3/2} \|\nabla_{\boldsymbol{u}} g\|_1^{1/2} \int_0^1 \sqrt{t} \, dt$$
$$= \frac{8\sqrt{2\alpha}}{3} \|\boldsymbol{d_u}\|_\infty^{3/2} \|\nabla_{\boldsymbol{u}} g\|_1^{1/2}$$
$$\le \sqrt{\alpha} 4 \|\boldsymbol{d_u}\|_\infty^{3/2} \|\nabla_{\boldsymbol{u}} g\|_1^{1/2}$$
$$= \sqrt{\alpha} 4 \left\|F_{\boldsymbol{r}}(\rho)^{-1}(\boldsymbol{e_u}(\rho) - \nabla_{\boldsymbol{u}} g)\right\|_\infty^{3/2} \|\nabla_{\boldsymbol{u}} g\|_1^{1/2}$$
$$\le \sqrt{\alpha} 12 \left\|F_{\boldsymbol{r}}(\rho)^{-1}\right\|_{1,\infty}^{3/2} \|\nabla_{\boldsymbol{u}} g\|_1^2 \quad \text{(Since Alg. 2 ensures } \|\boldsymbol{e_u}(\rho)\|_1 \le \frac{\eta \|\nabla_{\boldsymbol{u}} g\|_1}{4})$$
$$= \sqrt{\alpha} O(\|\nabla_{\boldsymbol{u}} g\|_1^2).$$

Plugging the above into (40) yields the desired result (15). ∎

| Cost | Wall-clock time (s) | Adaptive | $\rho_0 = 0$ |
|---|---|---|---|
| $L_1$ dist. | Median | 2.09 | 6.26 |
| | 90th %ile | 3.12 | 12.07 |
| | 10th %ile | 1.48 | 4.70 |
| $L_2^2$ dist. | Median | 5.49 | 25.74 |
| | 90th %ile | 10.18 | 48.27 |
| | 10th %ile | 1.40 | 4.56 |

Table 3: Comparison of median, 90th and 10th percentile performance for adaptively and naively initialized $\rho$ over 30 problems from the upsampled MNIST dataset with $L_1$ and $L_2^2$ costs ($\gamma_i = 2^4, \gamma_f = 2^{18}, p = 1.5$, and $q^{(1)} = 2$ initially).

## B  ADAPTIVE INITIALIZATION OF THE DISCOUNT FACTOR

Recall that in Alg. 2, we initialize the discount factor $\rho$ at 0 and anneal $(1 - \rho)$ by taking:

$$\rho \leftarrow 1 - (1 - \rho)/4$$

in L4 of the algorithm until the forcing inequality (6) is satisfied. Here, we describe a simple, practical strategy to reduce the overhead associated with this annealing procedure and the solving of a sequence of linear systems (see also the proof of Thm. 3.2 in Appx. A.1.4).

In particular, we initialize NewtonSolve (Alg. 2) with an initial guess $\rho_0$ (rather than 0) in practice. Each call to NewtonSolve returns the final discount factor $\rho$ found by the algorithm in addition to the descent direction $d_u$. Then, the next time NewtonSolve is called, we call it with

$$\rho_0^{\text{new}} = \max\left(0, 1 - (1 - \rho^{\text{old}}) * 4\right), \tag{42}$$

where $\rho^{\text{old}}$ is the discount factor returned by the previous NewtonSolve call. That is, the annealing starts from the second last annealing step of the previous call. As the linear system has changed since the previous call, this allows for a smaller discount factor to potentially replace the previously feasible one, if appropriate. We find that this simple change in the implementation improves performance empirically, as shown in Table 3. This version of the algorithm is used in the main experiments presented in Sec. 4.

## C  PROBLEM SIZE EXPERIMENTS

Here, we conduct experiments with varying problem size $n$ to empirically study the dependence of MDOT–TruncatedNewton on $n$. Fig. 4 shows the behavior over MNIST and color transfer problems with $L_1$ and $L_2^2$ cost functions, and problem size adjusted by down- or up-sampling the images. In all experiments here, we fix regularization weight at $\gamma_f = 2^{12}$. In addition to empirical behavior of MDOT-TruncatedNewton, we include a polynomial $f(n) = an^2$ passing through the empirical curve at the largest $n$; the curve explains the behavior of the algorithm well for large $n$. It performs no worse than $O(n^2)$ empirically for the problems considered in Fig. 2.

## D  ADDITIONAL BENCHMARKING ON DOTMARK

In this section, we extend the study in Fig. 2 with 10 more datasets from the DOTmark benchmark introduced by Schrieber et al. (2017) for benchmarking of discrete OT solvers. Schrieber et al. (2017) proposed 10 different image sets, *"to represent a wide range of theoretically different structures, while incorporating typical images that are used in praxis and/or have been used for previous performance tests in the literature"*. Example image sets include various kinds of randomly generated images, classical test images and real data from microscopy; each dataset consists of 10 grayscale images, yielding a total of 45 discrete OT problems, where the marginals $r, c$ are formed based on pixel values (Schrieber et al., 2017). The cost matrix is constructed similarly to the MNIST dataset from distances in 2D pixel locations. While Schrieber et al. (2017) proposed only to use the $L_2^2$ cost function, we evaluate on both $L_1$ and $L_2^2$ costs functions for consistency with Fig. 2 and for the sake of broader evaluation. Once again, for consistency with Fig. 2, we take $64 \times 64$ images, which yield $n = 4096$.

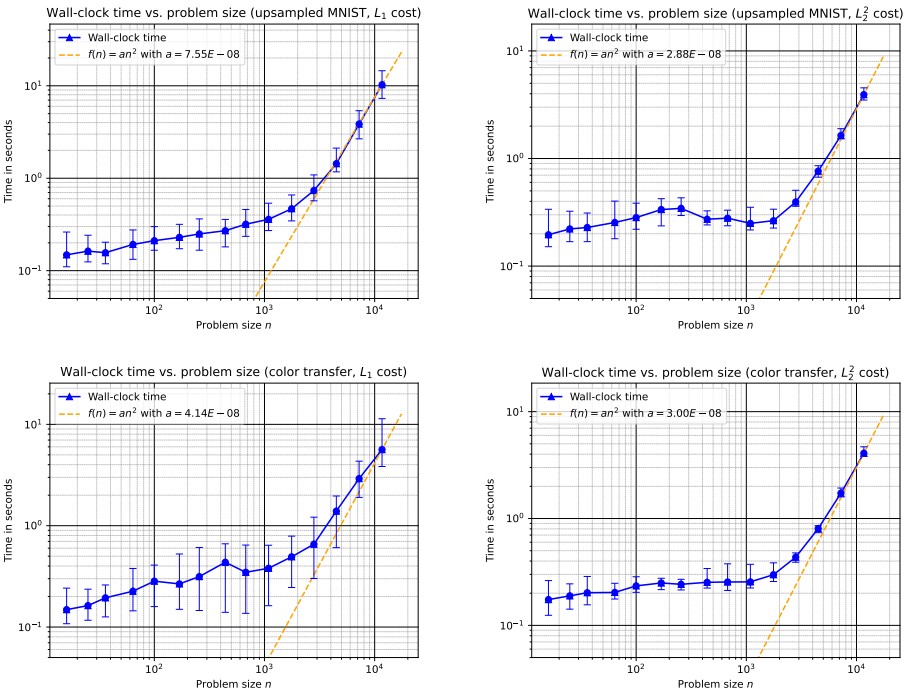

Figure 4: Log-log plot of wall-clock time for MDOT-TruncatedNewton vs. problem size $n$. Each marker shows the median over 60 random problems from the MNIST (top) and color transfer (bottom) problem sets with normalized $L_1$ **(left)** and $L_2^2$ **(right)** distance costs. Error bars show $10^{th}$ and $90^{th}$ percentiles. For all problems, $\gamma_f = 2^{12}$. Dashed lines show a polynomial $f(n) = an^2$, where $a$ is selected so that $an^2$ equals the median time taken at the largest $n$ considered. Above, the algorithm behaves no worse than $O(n^2)$.

For each of 20 problem sets (corresponding to a class of images and a cost function), we sample 20 random problems out of the 45 possible problems. Figs. 5-14 show the median time to converge for each algorithm at a given hyperparameter setting, and the error $\langle P - P^*, C \rangle$ after rounding the output of the algorithm onto $\mathcal{U}(\boldsymbol{r}, \boldsymbol{c})$ – with the exception of Alg. 3.5 of Feydy (2020); see Appx. E of Kemertas et al. (2025). The wall-clock time plots for the respective cost functions ($L_1$ and $L_2^2$) follow the same trends seen in the two datasets considered in Fig. 2. Following Kemertas et al. (2025), we include $75\%$ confidence intervals along both axes here, and also show that MDOT-TruncatedNewton is generally robust even at high precision, where maintaining numerical stability can be more challenging. Our conclusions based on Fig. 2 regarding the wall-clock time convergence behavior of MDOT-TruncatedNewton and how it compares to baselines remain unchanged.

**Operation Counts.** In addition to wall-clock time, we count here the number of primitive operations costing $O(n^2)$ for each algorithm. Examples of such *primitive* operations involving $n \times n$ matrices include row/column sums of matrices, matrix-vector products, element-wise multiplication of matrices, element-wise exponentiation/logarithm of matrices, addition/subtraction/multiplication/division between a matrix and a scalar, max over all entries of a matrix, summation over all entries of a matrix, etc. We count the number of primitive operations rather than a higher level function call such as the number of gradient evaluations due to inherent differences in the design of the various baseline algorithms; e.g., some methods require costly line search or inner loops between gradient evaluations. For all 20 problem sets displayed in Figs. 5-14, we find that the total number of $O(n^2)$ operations predict wall-clock time very well (high correlation), especially when the algorithms are run for long enough, as seen visually upon comparing top and bottom rows of the same column. All algorithms follow the same trend seen in Fig. 2, so that our conclusions once again remain the same.

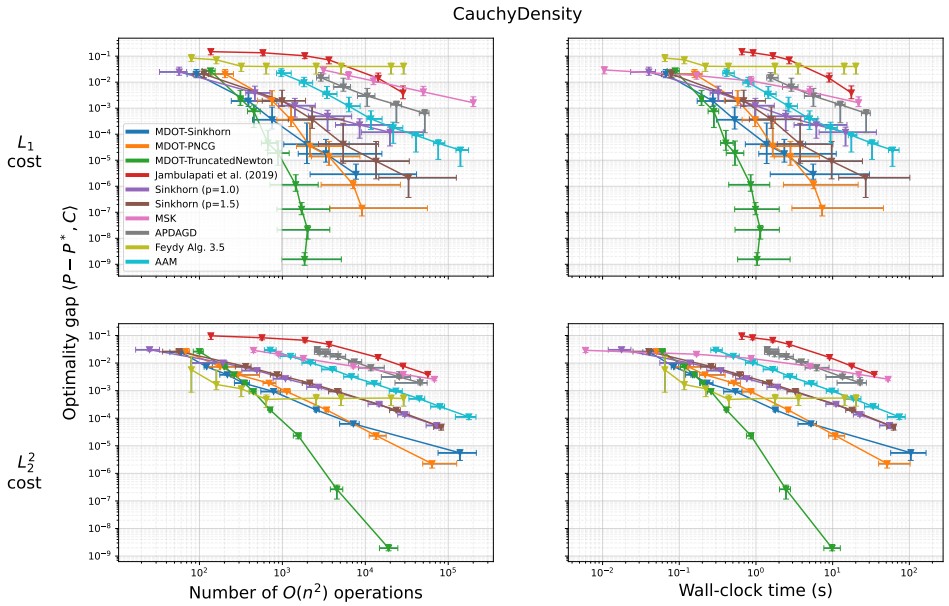

Figure 5: `CauchyDensity` problem with $L_1$ (top) and $L_2^2$ (bottom) costs, showing optimality gap vs. number of $O(n^2)$ operations (left) and wall-clock time (right).

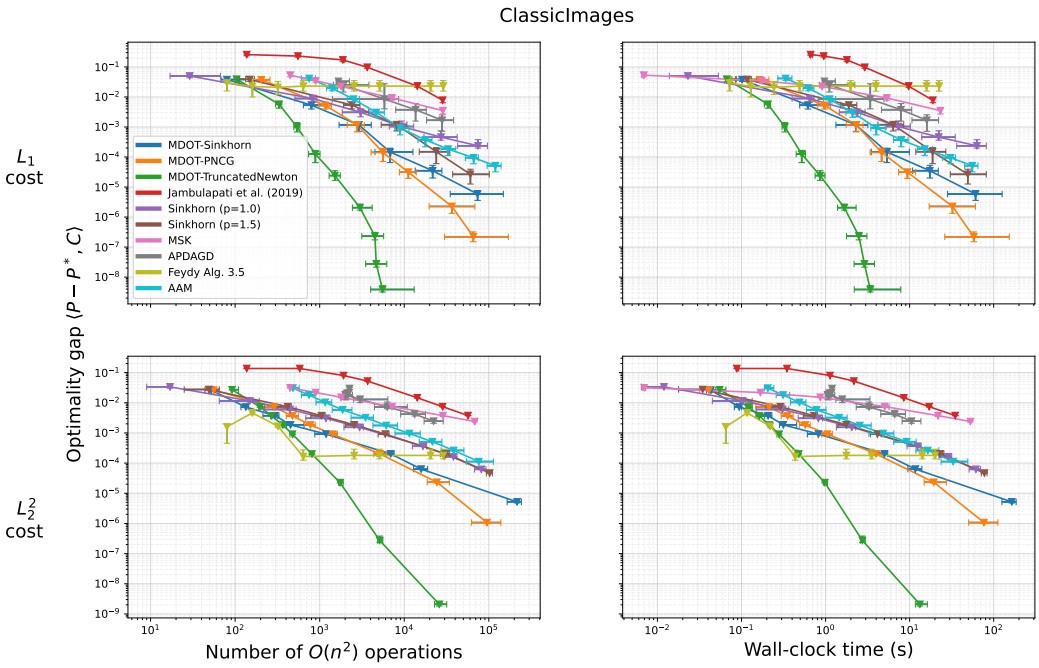

Figure 6: `ClassicImage` problem with $L_1$ (top) and $L_2^2$ (bottom) costs, showing optimality gap vs. number of $O(n^2)$ operations (left) and wall-clock time (right).

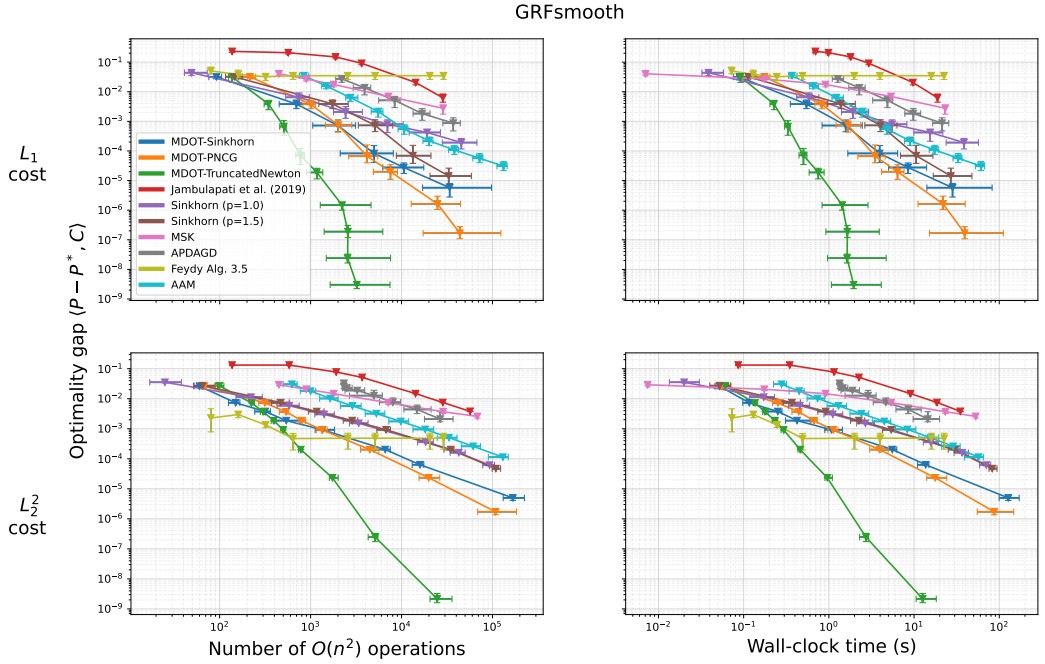

Figure 7: `GRFSmooth` problem with $L_1$ (top) and $L_2^2$ (bottom) costs, showing optimality gap vs. number of $O(n^2)$ operations (left) and wall-clock time (right).

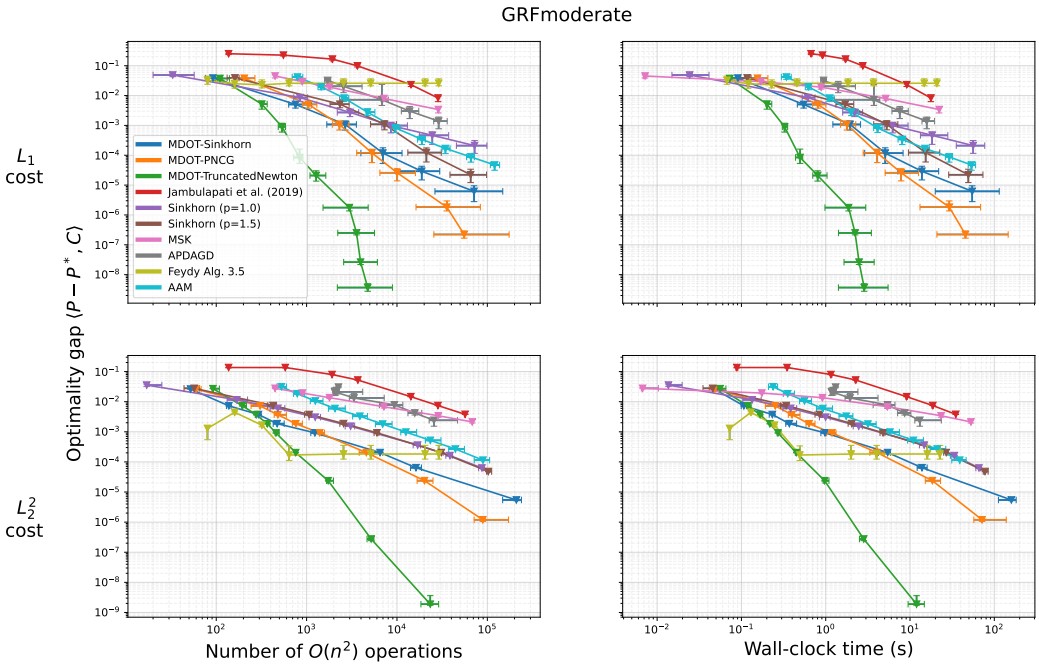

Figure 8: `GRFModerate` problem with $L_1$ (top) and $L_2^2$ (bottom) costs, showing optimality gap vs. number of $O(n^2)$ operations (left) and wall-clock time (right).

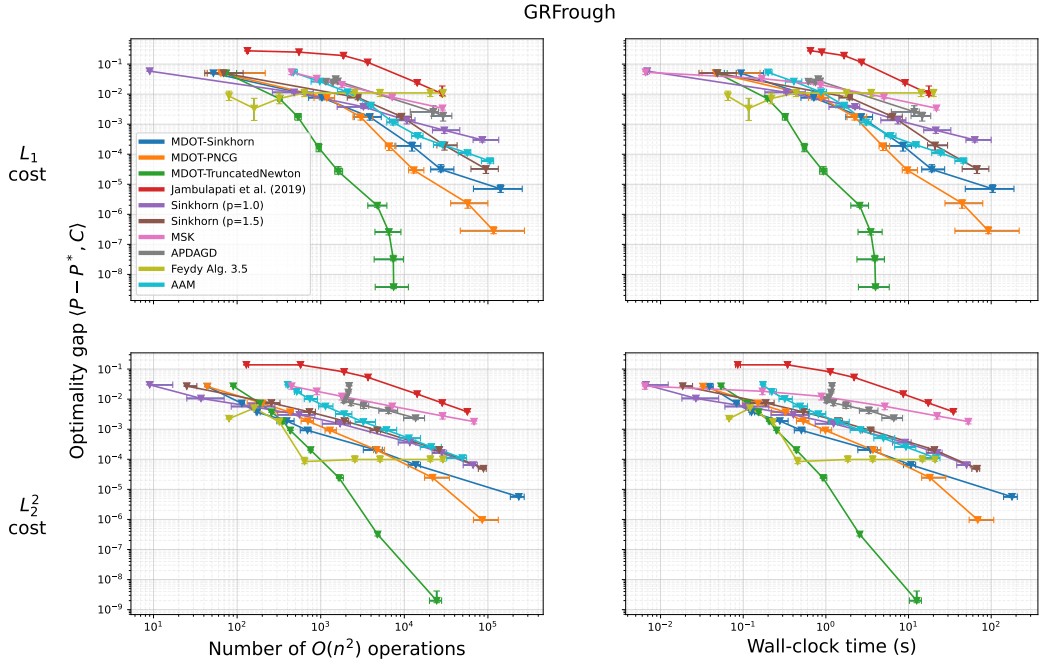

Figure 9: `GRFRough` problem with $L_1$ (top) and $L_2^2$ (bottom) costs, showing optimality gap vs. number of $O(n^2)$ operations (left) and wall-clock time (right).

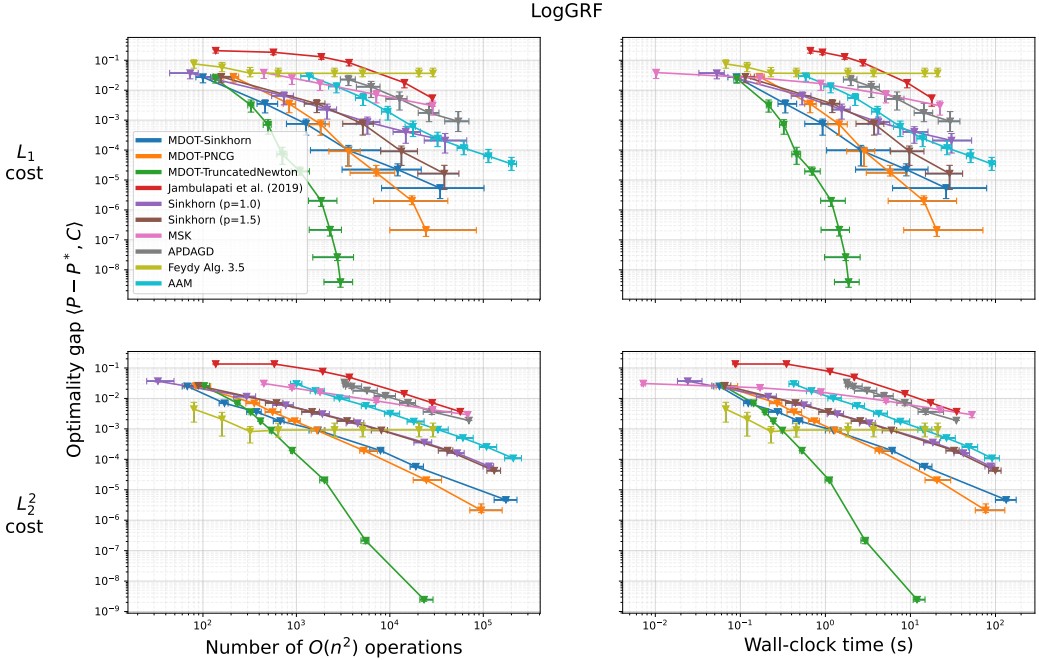

Figure 10: `LogGRF` problem with $L_1$ (top) and $L_2^2$ (bottom) costs, showing optimality gap vs. number of $O(n^2)$ operations (left) and wall-clock time (right).

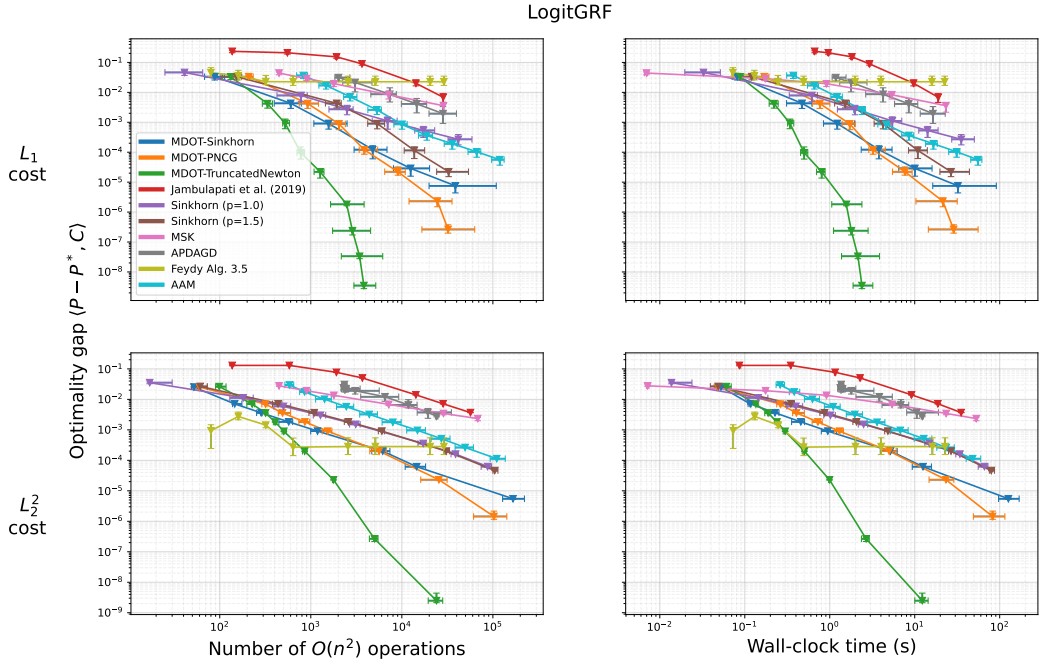

Figure 11: `LogitGRF` problem with $L_1$ (top) and $L_2^2$ (bottom) costs, showing optimality gap vs. number of $O(n^2)$ operations (left) and wall-clock time (right).

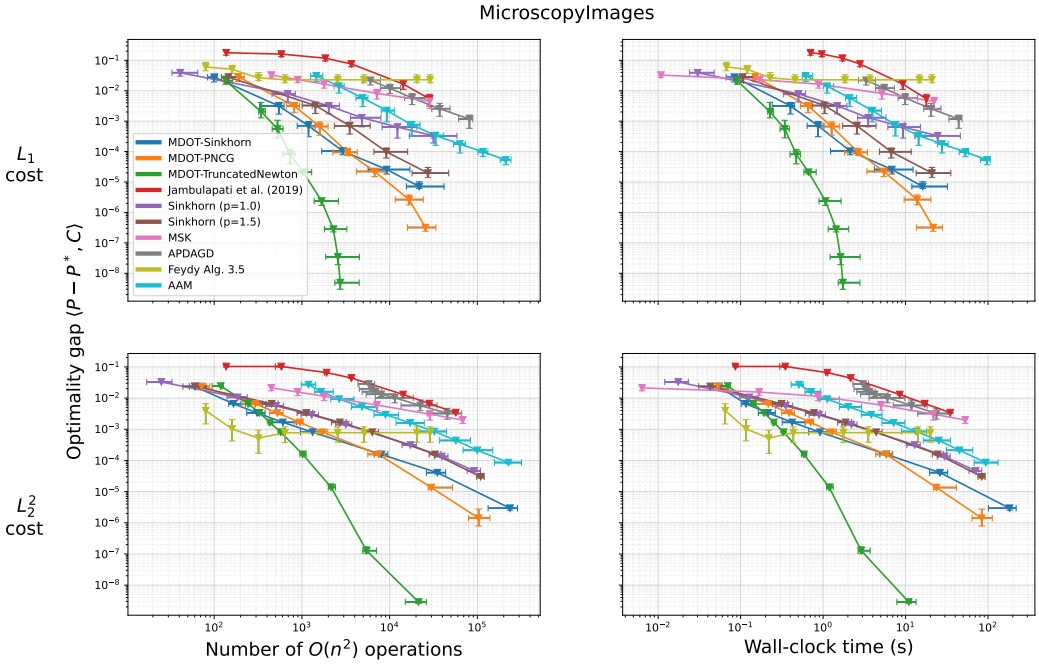

Figure 12: `MicroscopyImage` problem with $L_1$ (top) and $L_2^2$ (bottom) costs, showing optimality gap vs. number of $O(n^2)$ operations (left) and wall-clock time (right).

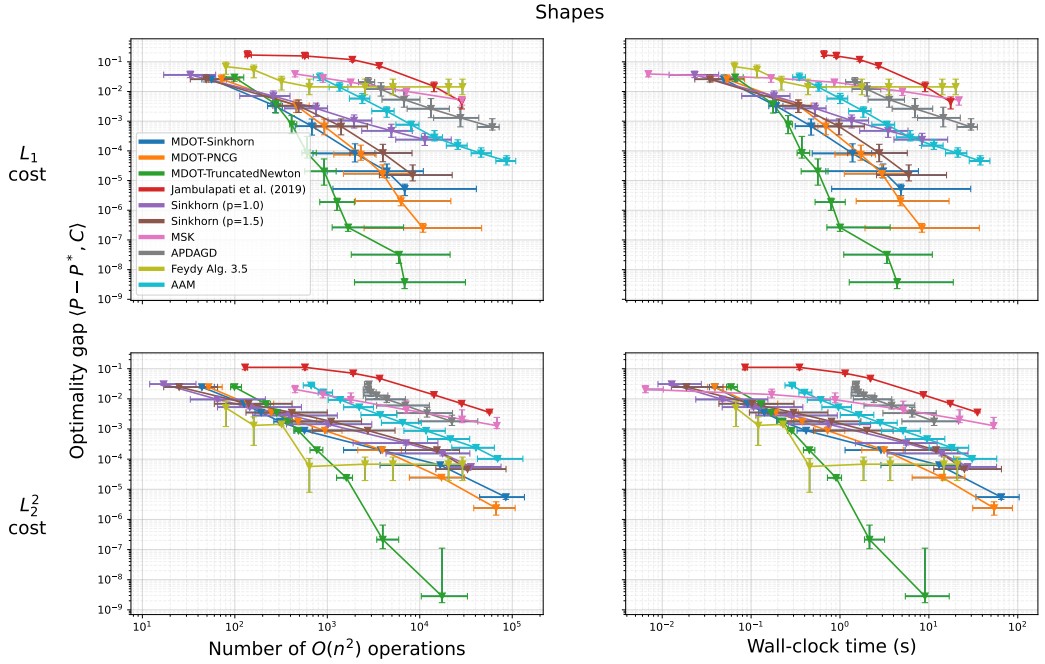

Figure 13: Shape problem with $L_1$ (top) and $L_2^2$ (bottom) costs, showing optimality gap vs. number of $O(n^2)$ operations (left) and wall-clock time (right).

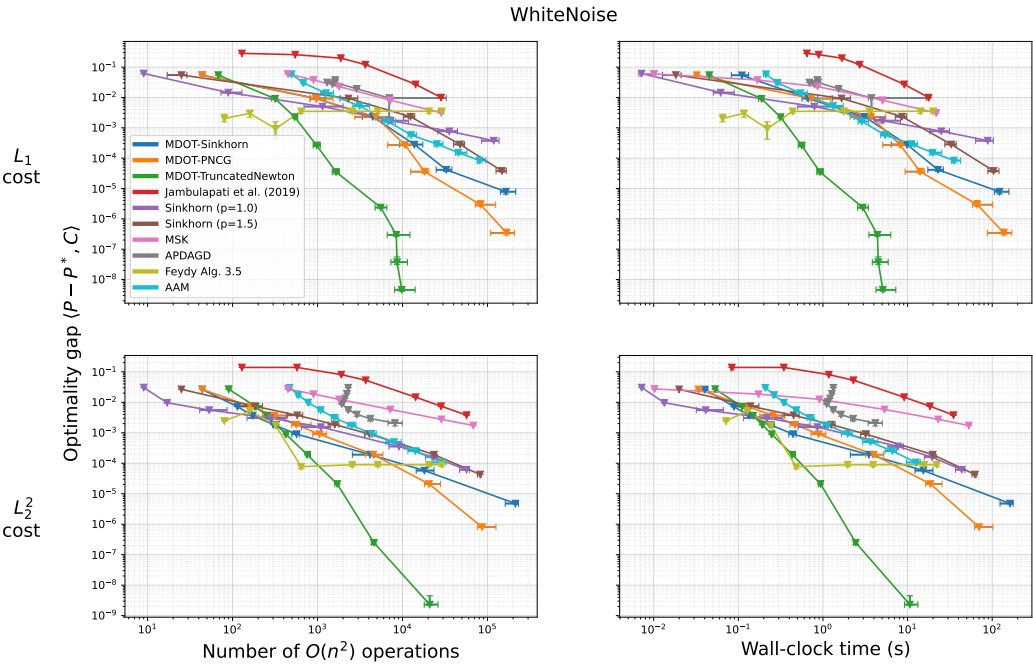

Figure 14: WhiteNoise problem with $L_1$ (top) and $L_2^2$ (bottom) costs, showing optimality gap vs. number of $O(n^2)$ operations (left) and wall-clock time (right).

# E ON THE SPECTRAL GAP $1 - \lambda_2$ AND DISCOUNTING

In Section 3, we demonstrated that the convergence of Algorithm 2, as established in Theorem 3.2 (and subsequently in Corollary 3.4), exhibits a worst-case dependence of $O((1-\lambda_2)^{-1/2})$ on the spectral gap $1 - \lambda_2$. In this section, we first explain why this dependence may be overly pessimistic. We then illustrate this observation through the experiments presented in Fig. 16.

Recall from Lemma A.6 that the convergence of CG for solving a $\rho$-discounted Newton/Bellman system $\boldsymbol{d_u} = \boldsymbol{s_{uv}} + \rho P_{\boldsymbol{rc}} \boldsymbol{d_u}$ is in fact $O((1-\rho)^{-1/2})$. In Thm. 3.2 we expressed the overall complexity of Alg. 2 in terms of $\lambda_2$ using the fact that the largest $\rho$ found by Alg. 2 via annealing satisfies $(1-\rho)^{-1} = O(1-\lambda_2)^{-1}$ in the worst-case; see (19) and the proof of Thm. 3.2. However, $1 - \rho$ tends to be much better behaved in practice and effectively mitigates this worst-case dependence on $1 - \lambda_2$. We introduce the rationale for this behavior via an example.

Suppose, given $n = 4$ the stochastic matrix $P_{\boldsymbol{rc}} \in \mathbb{R}^{4 \times 4}_{>0}$ has the form:

$$P_{\boldsymbol{rc}} = Q + \Delta, \quad Q := \begin{bmatrix} 0.75 & 0.25 & 0 & 0 \\ 0.25 & 0.75 & 0 & 0 \\ 0 & 0 & 0.75 & 0.25 \\ 0 & 0 & 0.25 & 0.75 \end{bmatrix},$$

where $\Delta \in \mathbb{R}^{4 \times 4}$ is an appropriately selected, tiny perturbation matrix with infinitesimal entries, and the Markov process given by $P_{\boldsymbol{rc}}$ can be (approximately) decoupled into two separate processes corresponding to the top-left and bottom-right blocks; the spectral gap of $P_{\boldsymbol{rc}}$ is tiny.

Consider also, for convenience, a stationary distribution $\boldsymbol{r}(P) = \begin{bmatrix} 0.25 & 0.25 & 0.25 & 0.25 \end{bmatrix}^\top$, and an initial distribution $\boldsymbol{r} = \begin{bmatrix} 0.5 & 0 & 0 & 0.5 \end{bmatrix}^\top$, which already assigns to each partition the correct amount of total mass as the stationary distribution $\boldsymbol{r}(P)$. For such a pair, $\boldsymbol{r}(P) - \boldsymbol{r}$ would be orthogonal to the first two eigenvectors $\boldsymbol{\nu}_1 = \begin{bmatrix} 1 & 1 & 0 & 0 \end{bmatrix}^\top$ and $\boldsymbol{\nu}_2 = \begin{bmatrix} 0 & 0 & 1 & 1 \end{bmatrix}^\top$ of $Q$. We can then easily show that the convergence of $\boldsymbol{r}$ to $\boldsymbol{r}(P)$ via repeated application of $P_{\boldsymbol{rc}}$ would be governed by $(1 - \lambda_3)^{-1} \approx 2$ rather than $(1 - \lambda_2)^{-1} = O(\|\Delta\|^{-1})$. Indeed, we can write:

$$Q = \boldsymbol{\nu}_1 \boldsymbol{\nu}_1^\top + \boldsymbol{\nu}_2 \boldsymbol{\nu}_2^\top + 0.5 \boldsymbol{\nu}_3 \boldsymbol{\nu}_3^\top + 0.5 \boldsymbol{\nu}_4 \boldsymbol{\nu}_4^\top,$$

which implies (with simple calculations):

$$\left\| \boldsymbol{r}(P)^\top - \boldsymbol{r}^\top P_{\boldsymbol{rc}}^l \right\|_1 = \left\| (\boldsymbol{r}(P) - \boldsymbol{r})^\top P_{\boldsymbol{rc}}^l \right\|_1 \leq \left\| (\boldsymbol{r}(P) - \boldsymbol{r})^\top Q^l \right\|_1 + O(\|\Delta\|_1 \|(\boldsymbol{r}(P) - \boldsymbol{r})\|_1)$$
$$= 0.5^l \chi(\boldsymbol{r} | \boldsymbol{r}(P)) + O(\|\Delta\|_1 \|\boldsymbol{r}(P) - \boldsymbol{r}(P)\|_1).$$

The idea here applies more generally to multiple eigenvalues near one, i.e., when the Markov process may be approximately separated into a larger number of groups. Since we call Alg. 2: NewtonSolve near the solution, where $\|\nabla g\|_1 = \|\boldsymbol{r}(P) - \boldsymbol{r}\|_1$ is small, the gradient (i.e., the RHS of the Newton system) tends to be nearly orthogonal to the subspace corresponding to near-zero eigenvalues of $I - P_{\boldsymbol{rc}}$. Consequently, the Newton/Bellman system can be discounted more aggressively than the worst-case suggested by the condition in (19), ultimately yielding $(1 - \rho)^{-1} \ll (1 - \lambda_2)^{-1}$.

In Fig. 16, we demonstrate this effect via experiments over 4 datasets from the DOTmark benchmark (Schrieber et al., 2017), where we display $(1-\rho)^{-1/2}, (1-\lambda_2)^{-1/2}$ and the number of CG iterations until convergence in the same plot. Since $P$ tends to get sparser with increasing $\gamma$ (Tang et al., 2024), we generally observe a decreasing spectral gap of $P_{\boldsymbol{rc}}$ with higher $\gamma$ (across all problem sets) as communication between particle groups is increasingly bottlenecked. In all cases, $(1-\rho)^{-1/2}$ predicts the total number of CG iterations better than $(1-\lambda_2)^{-1/2}$. Especially for the Shapes problem, the marginals

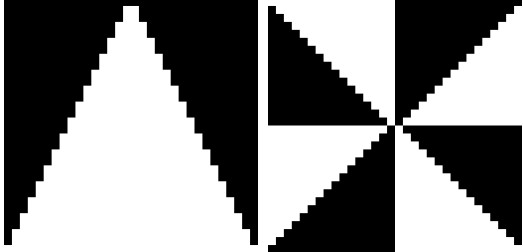

Figure 15: Sample images from Shapes.

$\boldsymbol{r}, \boldsymbol{c}$ contain lots of near-zeros and often yield non-communicating particle groups. Even though $(1-\lambda_2)^{-1/2}$ explodes quickly in this case, the method remains robust and $(1-\rho)^{-1/2}$ continues to be predictive of good performance.

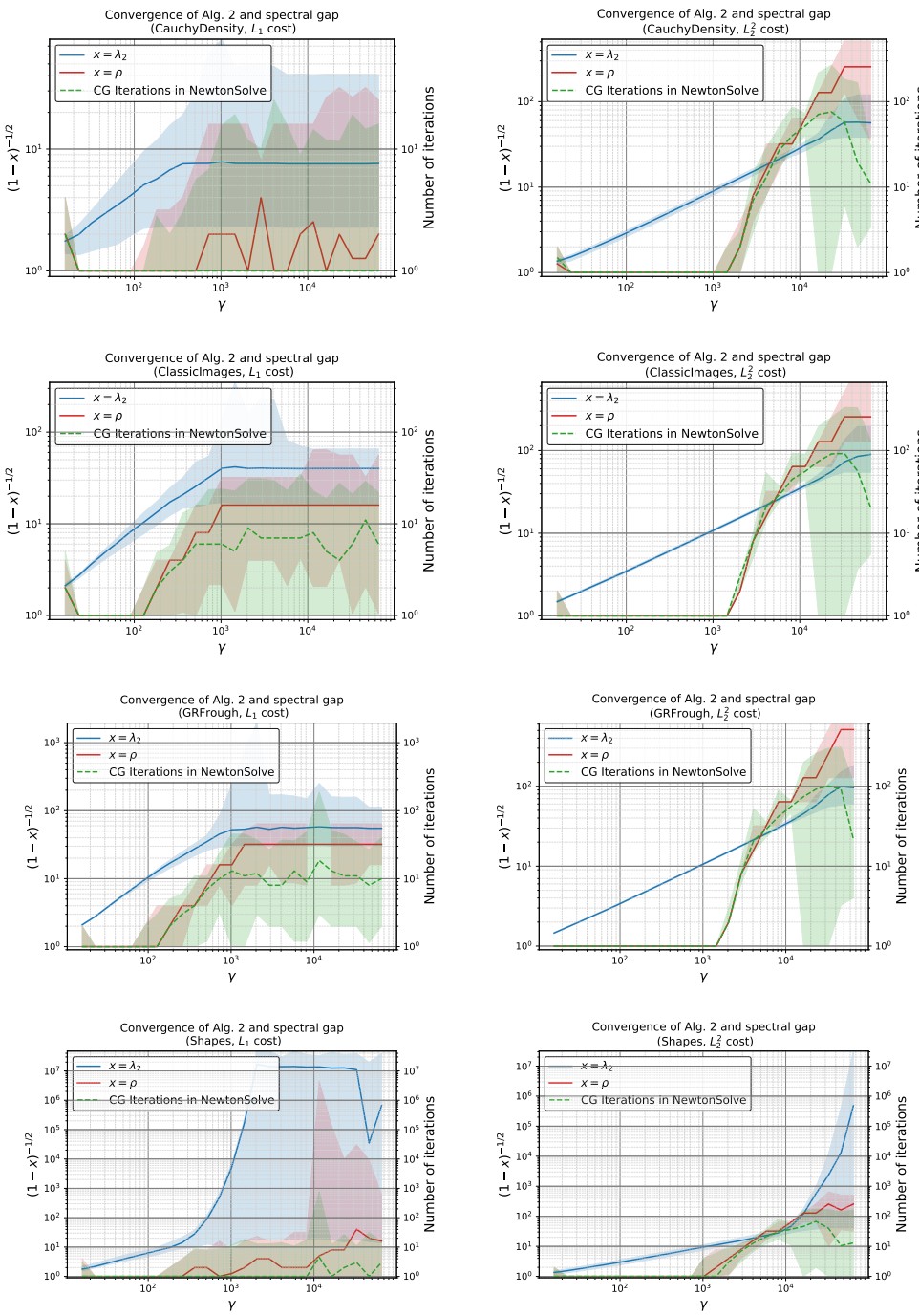

Figure 16: Over 4 tasks from DOTmark ($n = 1024$) with $L_1$ (**left**) and $L_2^2$ (**right**) costs, we run MDOT-TruncatedNewton with a fixed temperature decay rate of $q = 2^{1/16}$ and compute the spectral gap $1 - \lambda_2$ explicitly via eigenvalue decomposition every-time Alg. 2: NewtonSolve is called (at fixed intervals of $\gamma$ on a log-scale). On the LHS of the $y$-axis, the $(1 - \lambda_2)^{-1/2}$ term seen in Thm. 3.2 and Cor. 3.4 and $(1 - \rho)^{-1/2}$ are displayed, where $\rho$ is the discount factor found by Alg. 2. On the RHS of the $y$-axis the total number of CG iterations taken by NewtonSolve is shown. While the $(1 - \lambda_2)^{-1/2}$ term shown in Sec. 3 may be adversely affected by extremely large values (see bottom row), the practical rate dictated by $1 - \rho$ is much better and more accurately describes the convergence of the algorithm. Lines show the median over all 45 problems and shaded areas show 90% confidence intervals. See Appx. E for further discussion.

