# OpenReview forum: "A Truncated Newton Method for Optimal Transport"
_ICLR.cc/2025/Conference — ICLR 2025 Poster_

### Official Review · Reviewer_4av9 · 2024-11-04

**Soundness:** 2
**Presentation:** 2
**Contribution:** 2
**Rating:** 5
**Confidence:** 3

**Summary:**

The paper presents a new Newton-based optimization method for entropic regularised optimal transport. The authors present a new three-stage method that combines truncated Newton method, conjugate gradient and mirror descent. Finally, the authors present experiments with fast convergence of the proposed method.

**Strengths:**

In the proposed framework, the authors propose a lot of engineering improvements to make the method more stable and effiicient from the practical point of view.

**Weaknesses:**

The Weaknesses and Questions are together.

1. The section 2.1. is concerning and confusing.  First of all, lines 70-92 are almost exactly copied from  Kemertas et al. (2023), which is concerning. Secondly, the problem 2 is wrongly defined. It is not convex. It should be $+H(P)$. Thirdly, The dual problem is also unusual for me. Normally, the dual problem has a form of LogSumExp. In the paper, it is the sum of exponents. Please, provide how do you get formulas 3 and 4 in Appendix.

2. The major results from my perspective are more focused on practice and involve a lot of engineering improvements. Hence, the major proof of success should be experiments. However, from my perspective, the experiments are not good enough for a practical paper. a) First of all, there is only one figure with 2 examples, which is not enough. b) Secondly, in Fig.2, the proper loss should be a duality gap, not a primal function value. As the primal function value excludes constraints' violation and can be misleading. Also, the figure with feasibility may also help in Appendix, as well as for the dual loss. c) The presented results in Fig.2 are presented without statistical measures. It means that it was only one run, which can be quite stochastic, especially with respect to wall-clock time. d) All methods are starting from different points, which is very concerning. e) The results are only presented with respect to wall-clock time. I would recommend adding another measure that is not based on code performance, for example number of calculated gradients or something like that. Otherwise, MDOT code can be optimized for performance and compete with other methods that are not optimized. f) Finally, I didn’t find any attached code to verify the performance.

3. Do I understand correctly, that at the beginning, in practice, there is a part of pure adaptive mirror descent to find a near solution as a starting point, and then the algorithm performs Algorithm 4?

**Questions:**

see weaknesses

---

> ### Author Response · Authors · 2024-11-19
> **Thank you for your feedback. Response (1/n)**
>
> We thank the reviewer for their time and constructive feedback that has helped strengthen the paper in the Rebuttal Revision. We clarify several misunderstandings, and discuss minor fixes and newly added results in the Appendix of the Rebuttal Revision below.
>
> **Similarity with prior work in the Background section.** We acknowledge the overlap noted by the reviewer, which reflects the shared context of our matching problem setup. In the Rebuttal Revision we have revised the wording and flow to reduce repetition. Note that lines 70–92 are purely expository and introduce no new information.
>
> **Convexity of problem (2):** We apologize for a missing minus sign in the definition of Shannon entropy in line 66, which has been fixed in the Rebuttal Revision. We suspect the reviewer’s confusion about convexity and $-H(P)$ stems from that typo. Since Shannon entropy is concave in $P$, the negative entropy is convex, so that problem (2) is convex.
>
> **Provide the derivation of the sum-of-exponentials form for the dual (eqn’s (3) and (4)) in the Appendix.** As suggested, we have provided a full derivation of sum-of-exponentials dual form (and its relationship to the log-sum-exp form) in Appx. 1.2 of the Rebuttal Revision. Note that both forms of the dual widely appear in the literature (see references in line 772 of the Rebuttal Revision; our form sum-of-exponents appears in Altschuler et al. (2017), Dvurechensky et al. (2018), Lin et al. (2019)).
>
> **A lot of engineering improvements:** We argue that our paper bridges the spectrum between “a collection of engineering improvements” and “an integration of mathematically rigorous ideas.” On the theoretical side, we provide approximately 10 pages of detailed proofs on local convergence rate analysis and the per-iteration cost of the overall algorithm, the computational complexity of key subroutines, and novel connections (to our knowledge) to the Bellman equations in reinforcement learning—connections that at least one reviewer (9Fpa) found compelling, in addition to the rest of the theoretical results. Our theoretical framework using the theory of convergence of stochastic matrices for analyzing a 2nd order method in computational EOT is novel to our knowledge and, as we show, leads to a substantial boost in performance. These contributions go beyond a purely practice-focused approach and aim to advance both understanding and application.
>
> **Only two examples in Fig. 2:** It is well-established that the cost function plays a critical role in governing convergence behavior. In Fig. 2, we tested on both $L_1$​ and $L_2^2$​ costs across two different tasks: upsampled MNIST and color transfer. This resulted in **four distinct problem sets**. Still, **we added Figs. 5-14 to the Appendix showing the same exact trend as Fig. 2 across 10 different tasks** (both with $L_1$ and $L_2^2$ costs), adding 20 more problem sets (see main comment above to all reviewers).
>
> Note also; our study benchmarks against **9 baseline algorithms** and transparently lists 3 failed baselines due to slow convergence. This level of comprehensive benchmarking goes beyond what is typically seen in work published recently at similar venues; e.g., Ballu & Berthet (2023), Tang et al. (2024).
>
> **Constraint violation not displayed in the y-axis of Fig. 2:** For all algorithms (with one exception), Fig. 2 displays the excess cost $\langle P - P^*, C \rangle$ after rounding onto $U(\mathbf{r}, \mathbf{c})$ via Alg. 2 of Altschuler et al. (2017), ensuring there is no constraint violation (within 64-bit machine precision). In other words, constraint violations are already accounted for by the excess cost displayed in Fig. 2 thanks to Lemma 7 of Altschuler et al. (2017) and Hölder’s inequality. The use of rounding is now clarified in the caption of Fig. 2.
>
> The only exception to the above is Alg. 3.5 of Feydy (2020), which performs poorly when run with “no-debiasing” followed by rounding. To give Feydy’s algorithm the best possible representation, we instead display its absolute error of the estimated cost under “debiasing” and without rounding, even though said approach does not return a feasible transport plan. This was described in Appx. D.

---

> > ### Author Response · Authors · 2024-11-19
> > **Response (2/2)**
> >
> > **“The presented results in Fig. 2 are presented without statistical measures. It means that it was only one run, which can be quite stochastic.”** As stated in the caption of Fig. 2 (line 453), each marker represents the median over 18 runs (at a given hyperparameter setting). Similarly, all other figures and tables in the paper report medians over at least 18 runs, ensuring robustness and consistency in the results. The only exception is Fig. 3, which is explicitly presented as a one-off demonstration of the algorithm’s scalability to $n \approx 10^6$ and is thoroughly discussed as such.
> >
> > See also Figs. 4-14 in the Appx. of the Rebuttal Revision, which show confidence intervals. **Results show MDOT-TruncatedNewton is robust**; error bars were not included in Fig. 2 simply to avoid overcrowding given the large number of baseline methods.
> >
> > **The baseline algorithms start from different points** due to differences in their design, initialization procedures, and prescribed hyperparameter settings, as specified by their respective authors. To clarify, each marker in Fig. 2 represents the median error and median wall-clock time obtained by running each algorithm to termination for a specific hyperparameter setting (revised caption in Rebuttal Revision to clarify). The leftmost marker for a given algorithm does not correspond to iteration 0, but rather to the algorithm's output for the hyperparameter setting that yields the lowest precision (e.g., Sinkhorn at a high temperature). For detailed hyperparameter settings, please refer to Appx. D.
> >
> > Hence, we believe this does not undermine the validity of the benchmarking, as the focus is on overall convergence behavior and performance in practical use-case scenarios.
> >
> > **Wall-clock time measurement only.** The algorithms use different units of computation, making measures like the number of calculated gradients not directly comparable. For example, some methods run complicated/costly line searches between gradient evaluations. While we argue that for this range of algorithms wall-clock time remains the most consistent performance measure, we agree that wall-clock time can be influenced by implementation details.
> >
> > To address concerns, for the DOTmark experiments (20 problem sets) added in Figs. 5-14, we added counters to each of 10 baseline algorithms for the total number of $O(n^2)$-costing primitives (element-wise multiplication of matrices, matrix-vector product, row/column sum or element-wise exponentiation of matrices, etc.) and showed side by side with wall-clock time. **Figs. 5-14 show that the total operation count correlates very strongly with wall-clock time across methods.**
> >
> > **Code will be open-sourced for practical use and reproducibility purposes.**
> >
> > **Is pure mirror descent first applied to find an approximate solution and then Alg. 4 used?** The overall algorithm runs Alg. 1 (mirror descent) as its outer loop and Alg. 4 (truncated Newton) in L7 of Alg. 1 as the inner loop, which together approximately solve the EOT problems at a decreasing sequence of temperatures $1/\gamma^{(t)}$.  The _adaptive_ mirror descent idea in Sec. 3.3 describes how to update the $q$ parameter (temperature decay rate) in L8 of Alg. 1, so that the warm-starting of the dual variables in L10 of Alg. 1 is effective. Here, “effective warm-starting” means Alg. 4 is initialized in the next iteration (at the new temperature) near its super-linear local convergence zone. Fig. 1 empirically shows that the proposed heuristic is able to achieve that, as confirmed by the ratio of actual to theoretically predicted reduction in gradient norm (based on local convergence analysis of Thm. 3.5) being approximately 1.

---

> ### Comment · Reviewer_4av9 · 2024-11-21
>
> Dear Authors,
>
> Thank you for the detailed answers! I am trying to understand the details behind the dual problem, however it is still confusing for me. Next, I list my questions, regarding the dual problem. If we converge to the common ground, I would increase my score.
>
> 1.  Do you assume that $\sum_{I,j} P_{i,j}=1$ for the original primal problem?
>
> 2. I believe that it is not allowed to drop constant terms (line 720) in primal and dual optimization problems as it would break primal-dual properties and mismatch primal function values with dual function values. Otherwise, it should be written using argmin and argmax.
>
> 3. What is the benefit of sum-exp formulation over log-sum-exp formulation?
>
> 4. Could you kindly advise if I missed it, in what theorem do you prove the primal-dual convergence of the proposed algorithm meaning that the duality gap is getting smaller?
>
>  I am still reviewing the experimental part of the rebuttal and will return with it in the next answer.

---

> ### Author Response · Authors · 2024-11-22
> **Response to Questions 1-3**
>
> Dear Reviewer,
>
> Thank you for your follow-up and for engaging deeply with the dual problem. We appreciate your openness to revisiting your evaluation and address your questions below.
>
> **Q1**.  In the derivation of the dual in Appx. A.1.2, we assume that $\sum_{ij}P_{ij} = 1$ for the primal problem, as the feasible set $U(\mathbf{r}, \mathbf{c})$ is a subset of the simplex. Any matrix whose row and column sums match probability vectors must have its entries sum to 1. Consequently, adding the term $1 - \sum_{ij}P_{ij}$ to the primal objective (as in line 731) does not affect the objective value for any feasible $P$, nor does it alter the optimal solution.
>
> If we had not switched to the KL divergence with the uniform distribution (but continued with negative Shannon entropy), the Lagrangian in (17) would instead take the form $\gamma \langle P, C \rangle + \langle P, \log P \rangle + \langle \mathbf{u}, \mathbf{r} - \mathbf{r}(P) \rangle + \langle \mathbf{v}, \mathbf{c} - \mathbf{c}(P) \rangle$. Following the same steps as in our derivation, this would again lead to (3) after a reparametrization of the dual variables; see for example, derivation by Lin et al. (2019) in the second half of their Sec. 2.
>
> In Appx. A.1.2, we opted for the KL divergence route in our exposition because of the nice cancellation of constants from line 740 to 741, and the eliminated need to reparametrize $\mathbf{u}, \mathbf{v}$.
>
> **Q2**.  This is correct if we were using the duality gap between the dual and primal objectives. However, in our work, we measure performance using the excess cost $\langle P - P^*, C \rangle$, i.e., the suboptimality of the primal solution returned by the algorithm. For this, we require (and ensure) that the algorithm’s output $P$ is strictly feasible, which we achieve using the rounding procedure of Altschuler et al. (2017). Their Lemma 7 guarantees that the rounding procedure does not increase the cost of $P$ by more than the $L_1$ norm of the gradient of the dual.
>
> In the paper, we do not suggest that the optimal value of the objective in (2) equals the optimal value of the objective in (4). Rather, in line 85 we only say that Problem (4) can be solved instead of Problem (2). That is, if one solves Problem (4) to obtain $\mathbf{u}^*, \mathbf{v}^*$, then plugging these into the closed form in (3) yields the optimal solution $P^*_\gamma$ of Problem (2).
>
> Finally, note that the objective of the unconstrained Problem (4) is in fact scaled by a constant factor of $\gamma$ for convenience (as mentioned in brackets in line 733). Specifically, we have $\gamma \langle P^*_\gamma, C \rangle - H(P^*_\gamma) = -g(\mathbf{u}^*, \mathbf{v}^*)$.
>
> **Q3**.  For our work, the sum-of-exponentials formulation is advantageous as it leads to simpler expressions for the gradient and the Hessian matrix, significantly simplifying the derivations and exposition in Eqs. (7–12), as well as some of the proofs in the Appendix. However, we acknowledge that some authors prefer the log-sum-exp formulation. For example, Lin et al. transitioned from the sum-of-exponentials form in their ICML 2019 work to the log-sum-exp form in their JMLR work. In their Remark 2, Lin et al. (2022) comment on this transition; they note that the log-sum-exp form of the objective is known to be smooth, and leverage this smoothness property in their theoretical analysis.
>
> Note also that we can easily obtain the objective value for the log-sum-exp formulation by simply $L_1$-normalizing $P(\mathbf{u}, \mathbf{v})$ and re-evaluating (4). Let the normalizing constant $N(\mathbf{u}, \mathbf{v}) \coloneqq \sum_{ij} P_{ij}(\mathbf{u}, \mathbf{v})$ for any given $\mathbf{u}, \mathbf{v}$, where $P(\mathbf{u}, \mathbf{v})$ is given by (3). If we update $\mathbf{u}$ by a scalar shift as $\mathbf{u} - \log N(\mathbf{u}, \mathbf{v}) \mathbb{1}_n$ (or, equivalently update $\mathbf{v}$), the resulting $P$ lies on the simplex, and the new value of (4) matches the log-sum-exp objective value prior to updating $\mathbf{u}$.

---

> ### Author Response · Authors · 2024-11-22
> **Response to Q4**
>
> **Q4**. Thank you for your question. Our algorithm is not a primal-dual method in the sense of simultaneously minimizing the primal and dual objectives while explicitly monitoring the duality gap. Instead, we focus on minimizing the dual objective until the $L_1$ norm of its gradient is below a prescribed threshold $\varepsilon_{\mathrm{d}}$ (L4 of Alg. 1), then use the rounding procedure of Altschuler et al. (2017) to map the dual solution to a primal solution that is strictly feasible. The rounding procedure ensures that the cost $\langle P, C \rangle$ does not increase by more than the dual gradient norm, thus providing bounded suboptimality. This approach does not rely on directly proving convergence of the duality gap but instead ensures an approximate primal solution with guarantees on feasibility. This is in the same spirit as Alg. 1 of Altschuler et al. (2017) and Alg. 2 of Dvurechensky et al. (2018). See also our high-level overview in lines 150-163, where we discuss the outer loop Alg. 1.
>
> If this misunderstanding arose due to unclear phrasing in the manuscript, we are happy to revise the relevant sections to make this distinction clearer.
>
> For the convergence of unconstrained dual minimization, while we did not prove a global convergence result (which is highly non-trivial for 2nd order methods), we proved a quadratic local convergence result in Thm. 3.5 in terms of the $L_1$ norm of the dual gradient. This is amenable to the aforementioned rounding lemma of Altschuler et al. (2017), which is also expressed in terms of the $L_1$ norm of the dual gradient.
>
> Note that quadratic local convergence here implies that if we initialize sufficiently close to the solution so that step size $\alpha=1$ is admissible (no line search is necessary) and choose $\eta = ||g_k||$, then the gradient norm after a single truncated Newton update becomes $||g_{k+1}||_1 = O(||g_k||_1^2)$ thanks to Eq. (15); e.g., a decrease from order $10^{-3}$ to order $10^{-6}$. This is why the per-iteration computational cost of Alg. 4 in Cor. 3.4 is of importance; because if we can initialize close to the solution, we should expect Alg. 4 to terminate in a handful of iterations at worst and a single iteration at best. Fig. 1 and Table 1 show us that this is indeed possible with some intuitive heuristics (adapting how quickly we decay the temperature $1/\gamma$ during the execution of Alg. 1).

---

> ### Author Response · Authors · 2024-11-25
> **Following up on our responses**
>
> Dear Reviewer,
>
> Thank you again for your thoughtful questions and engagement with our paper. We have carefully addressed your concerns regarding the sum-of-exponentials form and related aspects in our rebuttal, providing detailed clarifications and adding new results to strengthen the manuscript. We hope these revisions address your concerns and provide the clarity needed to converge on common ground.
>
> As the end of the rebuttal period is approaching, we kindly ask if you might have the opportunity to review our responses and share any further thoughts or updates to your evaluation. If there are any remaining points of confusion or areas needing additional clarification, we would be happy to address them promptly or apply revisions.
>
> Thank you again for your time and valuable feedback.
>
> Best regards,
>
> Authors

---

> > ### Comment · Reviewer_4av9 · 2024-11-26
> >
> > Dear Authors,
> >
> > thank you for the detailed answer. Now, I have a better understanding of the proposed method and its theoretical convergence. I increased my score.
> >
> > Best regards, Reviewer.

---

> > > ### Author Response · Authors · 2024-11-27
> > >
> > > In their original review (under Weaknesses), the reviewer outlined three concerns under bullet point 1, six concerns labeled (a) to (f) under bullet point 2, and a question under bullet point 3. In our rebuttal, we believe we have addressed all nine concerns with detailed clarifications, minor revisions and additional experiments with convincing results, and we believe we have thoroughly answered the question.
> > >
> > > We kindly wish to understand if there are any remaining issues or aspects of the paper that the reviewer feels still need improvement, as the reviewer's current score (5/10) suggests it remains marginally below the acceptance threshold. If there are specific areas where further clarification would strengthen the paper, we would be happy to address them.

---

### Official Review · Reviewer_A2Da · 2024-11-04

**Soundness:** 2
**Presentation:** 2
**Contribution:** 2
**Rating:** 6
**Confidence:** 2

**Summary:**

The paper introduces a truncated newton method for solving entropic optimal transport problem. The algorithm is GPU-parallel and is scalable for large OT problem.

**Strengths:**

1. The proposed algorithm has good practical performance under GPU settings.
2. The author has done a good job in discussing the related work.

**Weaknesses:**

1. Please clarify the definitions of $\epsilon$ and $\epsilon^\prime$. Current presentations are confusing.
2. The proposed algorithm lacks a theoretical convergence analysis and a comprehensive assessment of its computational complexity. While popular OT solvers like Sinkhorn and APDAGD offer promising guarantees on total computational complexity, the theoretical complexity of this algorithm remains unclear to me.

**Questions:**

1. Could the author make the theorems self-contained? For example, what is $\lambda_2$ in Corollary 3.4?
2. What does helper Sinkhorn mean in Line 196?
3. The definition for LSE is not clear in Line 066-067.

---

> ### Author Response · Authors · 2024-11-19
> **Thank you for your feedback.**
>
> Thank you for appreciating the practical performance of our method and discussion of related work.
>
> **Regarding the lack of a general theoretical rate**, we acknowledge the lack of a complete global convergence rate theory for MDOT-TruncatedNewton; this was listed as an important future direction at the end of the paper. However, our efforts on _local_ convergence analysis have yielded useful insights and very strong empirical results (more than $10 \times$ improvement in speed at high precision in high dimensions) that suggest the possibility of rates better than the best known theoretical rate $\tilde{O}(n^2 \varepsilon^{-1})$, at least for certain OT problems including some 24 problem sets we have benchmarked on (see comment above directed to all reviewers and Figs. 4-14 in the Appendix of the Rebuttal Revision).
>
> **$\epsilon$ vs. $\varepsilon$ notation cleanup.** Among other improvements to the notation (listed under our response to 9Fpa), we changed the notation from $(\varepsilon, \epsilon, \epsilon^\prime)$ to $(\varepsilon, \varepsilon_\mathrm{d}, \varepsilon_{\chi})$ to improve clarity. Under the revised notation in the Rebuttal Revision:
> - $\varepsilon_\mathrm{d}$ is the tolerance on **d**ual gradient norm,
> - $\varepsilon_\chi$ is the threshold on $\chi^2$ divergence and Alg. 4 chooses $\varepsilon_\chi = \varepsilon_{\mathrm{d}}^{2/5}$ (clarified in line 299 of the Rebuttal Revision),
> - The variable $\tilde{\epsilon}$ has been removed entirely and Thm. 3.2 is now expressed in terms of $\varepsilon_{\mathrm{d}}$.
> - $\varepsilon$ remains an upper bound on the excess cost $\langle P - P^*, C \rangle$ for a feasible plan $P$ returned by an algorithm, in line with notation in prior work (Altschuler et al., 2017; Dvurechensky et al., 2018; Lin et al., 2019).
>
>
> **Response to Questions**
>
> **We revised Cor. 3.4 to be self-contained** ($\lambda_2$ is defined again) in the Rebuttal Revision.
>
> **The phrase “helper Sinkhorn routine”** referred to Alg. 3 - ChiSinkhorn. We revised the second sentence of Sec. 3: “In Section 3.1, we first develop a technique for obtaining the truncated Newton direction with convergence guarantees **(Alg. 2)** and introduce a helper ~~Sinkhorn~~ routine **(Alg. 3)** to improve the convergence rate of this algorithm.” The latter is called a “helper routine”, because it only serves to help control the convergence rate of the former in light of Thm. 3.2.
>
> **LogSumExp reductions** along rows and columns of a matrix $X$ have been defined explicitly and renamed to $\mathrm{LSE}_r(X)$ and $\mathrm{LSE}_c(X)$ respectively, to improve clarity: $\mathrm{LSE}_r(X) \coloneqq \log \big(\exp(X) \mathbb{1} \big)$ and $\mathrm{LSE}_c(X) \coloneqq \log \big(\exp(X^\top) \mathbb{1} \big)$.

---

> > ### Author Response · Authors · 2024-11-25
> > **Following up on our rebuttal**
> >
> > Dear Reviewer,
> >
> > We hope you’ve had a chance to review our responses to your concerns. As the rebuttal period is nearing its end, we kindly invite you to share any further thoughts or updates to your evaluation.
> >
> > We agree that a limitation of the current theoretical results is a lack of a global convergence rate proof.  However, we note that our method, like many second-order approaches (e.g., L-BFGS, an industry standard), emphasizes fast local convergence over global rates. As demonstrated in the empirical results (see Table 1 and Fig. 1) this rapid local convergence is leveraged with warm-starting and the annealing of temperature $1/\gamma$ to vastly improve empirical performance on all 24 benchmark problems (Fig. 2 and Figs. 5-14).
> >
> > We hope our clarifications and the new experimental results address your concerns, and we would be happy to provide further details if needed.
> >
> > Thank you for your time and thoughtful consideration.
> >
> > Best regards,
> > Authors

---

### Official Review · Reviewer_9Fpa · 2024-11-08

**Soundness:** 3
**Presentation:** 2
**Contribution:** 3
**Rating:** 6
**Confidence:** 4

**Summary:**

This paper introduces a truncated Newton method, a second-order algorithm, to solve the Entropic-regularized Optimal Transport problem.

**Strengths:**

The paper has interesting theoretical results, such as the connection to the Bellman equation (though I'm not sure if this is the first work to establish that connection) and the convergence in Chi-square distance of Sinkhorn algorithm (again, I'm not sure if this result is previously established and to the best of my knowledge this is the first time I see this result).

Theorem 3.2 is interesting since it's a log(1/eps) convergence result, though I skeptical of the dependence on other terms could harm the convergence. The authors should discuss this result, as to the best of my knowledge, the only other equivalent result is in this work [1] (UOT can be viewed as a generalization/approximation of OT, I think this work should be cited in discussion to this work as well).

The paper has strong experiment settings that show superior result on wall-clock time and the color transfer result is very nice. The authors also have a detailed description for an efficient implementation of proposed algorithm.

[1] "On Unbalanced Optimal Transport: Gradient Methods, Sparsity, and Approximation Error".
Quang Minh Nguyen, Hoang Huy Nguyen, Lam Minh Nguyen, Yi Zhou
Journal of Machine Learning Research, (JMLR), 2023

**Weaknesses:**

The complexity in Theorem 3.2 depends on the spectral gap of P_{rc}, but there is no control over this spectral gap. I suspect that this could lead to very poor convergence should the spectral gap is very bad. I recommend having some analysis on this term to make the results stronger. Additionally, it is possible for this term to be $n$ dependent too, so the $n$ dependence of the complexity might more than just $O(n^2)$. If that this is the case, the method is a tradeoff between $n$ and $\epsilon$.

The paper is a bit hard to read and it is hard to follow the intuition of the paper. The proof should have been explained better, such as the chi-square convergence. Furthermore, the notations are not clearly explained or presented (see below).

**Questions:**

Various notations used in the manuscript is not clear or hard to find the definition. For example, what is $\mu$? Some intuitive explanation of $P_{rc}$ would be nice also. I recommend adding more to the notation section and explain these better.

In the experiment section, can the authors compare the method with other second-order methods, if any?

Also, can you give an intuitive explanations of how you do the proofs (main ideas or a high-level analysis like an ODE analysis)? I'm very interested in the results, and maybe in the later iterations of the paper you can incorporate these proof sketches to help the readers also.

---

> ### Author Response · Authors · 2024-11-19
> **Thanks very much for your feedback.**
>
> We thank the reviewer for their feedback, and their appreciation of both our empirical and theoretical results.
>
> We first clarify a point in Strengths.
>
> **$\log(1/\epsilon)$ convergence of Alg. 2 in Thm. 3.2.** We would like to clarify that, the dependence on $\epsilon$ in Thm. 3.2, as the reviewer astutely noticed, is indeed possibly worse than $\log(1/\epsilon)$ due to other terms. In particular, we generally have $||\mathbf{r} - \mathbf{r}(P)||_1 \leq \chi(\mathbf{r} | \mathbf{r}(P))$; see Eq. 2.12 of Fill (1991). This is why we run ChiSinkhorn (Alg. 3) each time before Alg. 2 is called: to control the ratio $\chi(\mathbf{r} | \mathbf{r}(P)) / ||\mathbf{r} - \mathbf{r}(P)||_1$. The specific call we make to ChiSinkhorn in line 4 of Alg. 4 results in the $\tilde{O}(\epsilon^{-2/5})$ dependence seen in Corollary 3.4 (proof in Appx. A.1.5). This is now clarified in line 299 of the Rebuttal Revision.
>
> Now, we address the questions, before responding to weaknesses.
>
> **Intuition on $P_{\mathrm{rc}}$**.
>
> For technical intuition on the matrix $P_{\mathrm{rc}}$, we refer the reviewer to Lemma A.1 in the Appendix and its proof, where we show that $P_{\mathrm{rc}}$ is a reversible, irreducible, row-stochastic matrix whose second largest eigenvalue $\lambda_2$ is strictly less than one, all eigenvalues are real and stationary distribution is given by $\mathbf{r}(P)$.
>
> We have also added a less technical intuitive discussion of $P_{\mathrm{rc}}$ above Lemma A.1 in the Rebuttal Revision (lines 801-813) that the reviewer may find helpful. The reader is now referred to this discussion in the Rebuttal Revision of the main text, in a footnote immediately below Thm. 3.2.
>
> **Comparison with other second-order methods.** Unfortunately, we were unable to find GPU-parallel implementations of alternative second order methods that can compute feasible, high precision solutions to the OT problem. Available options we could find use scipy’s L-BFGS solver, which is CPU-based and substantially slower.
>
> **High level intuition on proofs.** The proofs rely heavily on the theory of convergence of stochastic matrices, techniques in the analysis of truncated Newton and linear CG methods in numerical optimization. Unfortunately, we had to defer proofs to the appendix due to space constraints with no room left in the main text even for sketches. For Thm. 3.2, the proof in the Appendix begins with a short high-level intuition; lines 1110-1115 in the Rebuttal Revision.
>
> Now, we discuss Weaknesses:
>
> **Lack of control over and precise characterization of the spectral gap $1-\lambda_2$. Possible poor dependence on $n$**
>
> Firstly, we agree with the reviewer that a detailed analysis on the spectral gap and its dependence on problem parameters (e.g., properties of the cost matrix and the marginals) is an interesting problem, but we suspect it requires extensive theoretical work beyond the scope of this paper.
>
> To alleviate the reviewers’ concerns around possible poor dependence on problem size $n$, we have revised Fig. 4 in the Appendix, which shows that the dependence on $n$ is no worse than $O(n^2)$, at least for the problems considered in the main text. We also remind the reviewer that Fig. 3 shows the solution of a color transfer problem at a staggering $n \approx 1$ million, which would not have been possible had the method generally suffered from a poor dependence on $n$. For the DOTmark experiments in the Rebuttal Revision (Figs. 5-14), while we have not explicitly ablated $n$, the runtimes at $n=4096$ largely agree with Fig. 2, so we expect similar trends in $n$. While it may be possible to construct edge cases where the spectral gap has poor dependence on $n$, the empirical results strongly suggest that for a wide variety of problems the method remains very robust.
>
> Lastly, we note that our analysis of the algorithm covers the worst-case convergence of diagonally-preconditioned CG based on the condition number of the preconditioned coefficient matrix of the _discounted Newton system_. However, a more precise characterization of CG convergence accounts for the entire spectrum of the matrix, and shows that it may converge much more quickly than otherwise predicted by the condition number if preconditioned eigenvalues are well-clustered on the real line (Thm. 5.5 of Nocedal & Wright, 2006); see discussion immediately after Thm. 3.2. We suspect the efficiency of our method may be partly owed to this behavior and believe this too is an important area of future research.

---

> ### Author Response · Authors · 2024-11-19
> **Response continued**
>
> **Notational clarity and ease of reading**. We hope that our efforts to improve readability have alleviated the reviewer’s concerns:
> - We included intuition on $P_{\mathrm{rc}}$ in A.1.3 and referred the reader to it in the main text.
> - We cleaned up the notation mixing $\varepsilon$ and $\epsilon$ to improve readability and clarity (details in our response to A2Da).
> - We note that $\mu$ was defined the first time it was used in the main text. It is the smallest diagonal entry of $P_{\mathrm{rc}}$. This is now repeated the second (and last) time it is mentioned in the main text.
> - We revised Cor. 3.4 to be self-contained ($\lambda_2$ is defined again).
> - We added derivations for Eqs. (3-4) in the Appendix upon request by reviewer 4av9.
> - LogSumExp reductions along rows and columns of a matrix $X$ have been defined explicitly in Sec. 2 (upon request by reviewer A2Da) and renamed to $\mathrm{LSE}_r(X)$ and $\mathrm{LSE}_c(X)$ respectively, to improve clarity.

---

> > ### Comment · Reviewer_9Fpa · 2024-11-23
> > **Reviewer response**
> >
> > I thank the authors for the detailed response. I was hoping for some characterization of the spectral gap so I will keep the score. Additionally, I'm hoping that the reviewer will include the epsilon dependence discussion as in the previous review and in the authors' response.
> >
> > Thank you very much.

---

> ### Author Response · Authors · 2024-11-28
> **On the spectral gap**
>
> Dear Reviewer,
>
> In the latest revision, we added a section Appx. F dedicated to the discussion of the spectral gap dependence seen in Thm. 3.2 and Cor. 3.4, as well as a more in-depth empirical analysis of the subject.
>
> First, we provide mathematical rationale (due to Lemma A.6) on why the $(1-\lambda_2)^{-1/2}$ rate can in fact be quite pessimistic, and $(1-\rho)^{-1/2}$ comprises a better bound, where $\rho$ is the discount factor found by Alg. 2 via annealing.
>
> Then, we motivate via an example why it is often the case that, in practice, we have $(1-\rho)^{-1} \ll (1-\lambda_2)^{-1}$ when $(1-\lambda_2)^{-1}$ is very large.
>
> Lastly, we conduct experiments shown in Fig. 16, validating that the convergence behavior of Alg. 2: NewtonSolve is indeed much better captured by $(1-\rho)^{-1/2}$. While $(1-\lambda_2)^{-1/2}$ may indeed explode in some cases, this theoretical dependence is effectively controlled/mitigated by our discounting framework, and is not seen to reflect negatively on the empirical performance of the algorithm (bottom row of Fig. 16).
>
> We thank you for your constructive feedback, which has helped us better present the inner workings of the proposed method.
>
> The pessimism of the spectral gap dependence relative to $(1-\rho)$ is now mentioned immediately after Thm. 3.2 in the main text (with added insight on the proof strategy), and the reader is directed to Appx. F for further discussion and empirical analysis.
>
> In line 290, we have also added mention of the possible dependence on $\varepsilon_d$ for the other terms appearing in (14).
>
> Sincerely,
>
> Authors

---

### Author Response · Authors · 2024-11-19
**To all reviewers**

We thank all reviewers for their questions, comments and valuable feedback.

In addition to individual responses below, we note the following;

**On error bars and dependence on problem size $n$:** We invite all reviewers to view the revised Fig. 4 in the Appendix of the Rebuttal Revision (page 27), where we include 10-90th percentile error bars for MDOT-TruncatedNewton over 60 randomly sampled problems, for each of all 4 problem sets in Fig. 2, and a range of $n$ values. Fig. 4 provides evidence that **empirical dependence of the algorithm on problem size $n$ is no worse than $O(n^2)$ for the 4 problem sets we considered in the main text**, up to at least $n \approx 10,000$. This is now highlighted in line 485 of the Rebuttal Revision (main text).

**On additional benchmarking over 10 more datasets $\times$ 2 cost functions, and inclusion of operation counts and error bars**. All reviewers are also invited to view the newly added **Figures 5-14 in the Appendix of the Rebuttal Revision** (page 28 onwards), where we run baseline comparisons similarly to Fig. 2 ($n=4096$) over 2 cost functions x 10 new datasets from the DOTmark benchmark of Schrieber et al. (2017). We display the median of 20 randomly sampled problems per problem set as well as 75% confidence intervals; **results show the same trends as Fig. 2**, adding to the evidence on the speed of MDOT-TruncatedNewton, as well as its astounding convergence behavior with respect to error $\varepsilon$, which far exceeds best known theoretical rates $O(\varepsilon^{-1})$. **Figs. 5-14 also include the number of $O(n^2)$-costing operation counts for each algorithm, which strongly correlate with runtime and show the same trends.** The reader is now pointed to these results in the beginning of Sec. 4 (lines 427-428).

---

### Meta-Review · Area_Chair_KRFa · 2024-12-20

**Metareview:**

This paper introduces a truncated Newton method, a second-order algorithm, to solve the Entropic-regularized Optimal Transport problem. The paper has strong experiment settings that show superior result on wall-clock time and the color transfer result is very nice. The authors also have a detailed description for an efficient implementation of proposed algorithm.

**Additional Comments On Reviewer Discussion:**

Reviewers had several concerns about the presentation of the paper. After the rebuttal I believe these are largely addressed.

---

### Decision · Program_Chairs · 2025-01-22

Accept (Poster)